

# Artificial intelligence and machine learning approaches in cerebral palsy diagnosis, prognosis, and management: a comprehensive review

Shalini Dhananjay Balgude[1,2], Shilpa Gite[1,2], Biswajeet Pradhan[3] and Chang-Wook Lee[4]

[1] Symbiosis Centre for Applied Artificial Intelligence (SCAAI), Symbiosis Institute of Technology, Symbiosis International (Deemed University) (SIU), Pune, Maharasthra, India
[2] AI & ML Department, Symbiosis Institute of Technology (Pune Campus), Symbiosis International Deemed University, Pune, Maharasthra, India
[3] Centre for Advanced Modelling and Geospatial Information Systems (CAMGIS), School of Civil and Environmental Engineering, Faculty of Engineering and IT, University of Technology Sydney, Sydney, New South Wales, Australia
[4] Department of Science Education, Kangwon National University, Chuncheon-si, Republic of South Korea

Corresponding authors
Shilpa Gite,
shilpa.gite@sitpune.edu.in
Chang-Wook Lee,
cwlee@kangwon.ac.kr

## ABSTRACT

Cerebral palsy (CP) is a group of disorders that alters patients' muscle coordination, posture, and movement, resulting in a wide range of deformities. Cerebral palsy can be caused by various factors, both prenatal and postnatal, such as infections or injuries that damage different parts of the brain. As brain plasticity is more prevalent during childhood, early detection can help take the necessary course of management and treatments that would significantly benefit patients by improving their quality of life. Currently, cerebral palsy patients receive regular physiotherapies, occupational therapies, speech therapies, and medications to deal with secondary abnormalities arising due to CP. Advancements in artificial intelligence (AI) and machine learning (ML) over the years have demonstrated the potential to improve the diagnosis, prognosis, and management of CP. This review article synthesizes existing research on AI and ML techniques applied to CP. It provides a comprehensive overview of the role of AI-ML in cerebral palsy, focusing on its applications, benefits, challenges, and future prospects. Through an extensive examination of existing literature, we explore various AI-ML approaches, including but not limited to assessment, diagnosis, treatment planning, and outcome prediction for cerebral palsy. Additionally, we address the ethical considerations, technical limitations, and barriers to the widespread adoption of AI-ML for CP patient care. By synthesizing current knowledge and identifying gaps in research, this review aims to guide future endeavors in harnessing AI-ML for optimizing outcomes and transforming care delivery in cerebral palsy rehabilitation.

## INTRODUCTION

Cerebral palsy (CP) encompasses a range of conditions affecting muscle coordination, posture, and movement, leading to various deformities in patients. It is an umbrella term that includes various symptoms in different individuals (*Sadowska, Sarecka-Hujar & Kopyta, 2020*). Cerebral palsy constitutes a cluster of enduring conditions affecting movement, posture, and motor function, arising from damage to brain tissues or abnormalities in the brain (*Bax et al., 2005*; *Rosenbaum et al., 2007*; *Cans et al., 2007*). The predominant symptoms of cerebral palsy involve motor function impairment, often co-occurring with sensory, cognitive, communication, epileptic, musculoskeletal deformities, and behavioral issues. Globally, an estimated 1.5 to three individuals per 1,000 live births are affected by cerebral palsy. The risk of cerebral palsy is influenced by several factors, such as congenital (by birth) malformations, restricted fetal growth, multiple pregnancies, infections during the fetal and neonatal stages, birth asphyxia (lack of oxygen), preterm birth, untreated maternal hypothyroidism, perinatal stroke, and thrombophilia (condition with increased tendency of blood to form clots) (*Stavsky et al., 2017*; *Hankins, 2003*). Among these, premature birth stands out as the primary cause of cerebral palsy (*Stavsky et al., 2017*; *Hankins, 2003*).

This comprehensive review focuses on implementing AI for cerebral palsy patient care, targeting researchers in the AI and machine learning field who are interested in conducting research in this area. The review provides an in-depth exploration of the current landscape of artificial intelligence (AI) and machine learning (ML) applications in the field of cerebral palsy (CP) care. It delves into the diverse AI methodologies employed in prior research, conducts thorough comparative assessments of various studies within this domain, and pinpoints the existing challenges, lacunae, and potential avenues for future research. By furnishing a broad overview of the current research landscape in this sphere, the review serves as an initial reference for researchers keen on pursuing further investigations in this realm to understand the problems in current methods and potential areas to work on. Ultimately, this groundwork could potentially pave the way for the integration of AI-driven, efficient, and cutting-edge solutions for CP diagnosis, prognosis, and personalized care into clinical practice.

CP exhibits various classifications based on factors such as the affected brain area, type of movement disorders, severity, and level of damage. Movement disorders are generally classified into three types: spastic, dyskinetic, and ataxic. Spastic syndrome, which arises from damage to the brain and neural pathways, affects movement control and can be further divided into monoparesis (weakness in one limb), hemiparesis (weakness affecting an arm and a leg on any one side of the body), triparesis (weakness in any three limbs), tetraparesis (weakness in all four limbs), and spastic diplegia—where the muscle tone of two limbs, usually legs is increased leading to stiffness in jery movements. Dyskinetic symptoms like jerky movements, writhing, and spasms occur due to injuries to subcortical structures (regions below the cerebral cortex). In contrast, ataxic symptoms characterizing poor muscle control and coordination are a consequence of cerebellar injuries. Spastic CP predominantly affects one or both sides of the body, with about 80% of cases exhibiting

increased muscle tone and reflexes. This category is further divided into unilateral or bilateral based on the extent of involvement. The classification based on affected limbs includes quadriplegic (involving all four limbs), hemiplegic (affecting one side of the body), diplegic (more pronounced in the legs than the arms), and monoplegic (affecting only one limb), with diplegic CP being the most prevalent form. Dyskinetic CP constitutes 10% to 20% of cases and is characterized by involuntary, uncontrolled, repetitive, and sometimes stereotypical movements, along with fluctuating muscle tone. Dystonic postures involve heightened muscle tone, while choreoathetosis refers to rapid, uncontrollable, twisting movements with reduced muscle tone. Ataxic CP, accounting for 5–10% of cases, manifests as coordination loss and hypotonia. Mixed CP, affecting approximately 15.4% of cases, results from damage across various brain regions, leading to a combination of two or more cerebral palsy types. Symptoms of mixed CP often include a blend of spastic and athetoid features (*Paul et al., 2022*).

The functional classification of cerebral palsy employs various systems, such as the Gross Motor Function Classification System (GMFCS), the Manual Ability Classification System (MACS), the Communication Function Classification System (CFCS), and the Eating and Drinking Ability Classification System (EDACS) (Fig. 1). GMFCS, introduced by Palisano et al. in 1997, is globally employed for assessing motor function in children aged 2–18 years (*Paul et al., 2022*; *Morgan et al., 2018*; *Arnaud et al., 2021*; *Al-Zwaini, 2018*). In the Gross Motor Function Classification System (GMFCS), Level 1 encompasses individuals capable of walking unaided. Level 2 includes individuals who can perform all activities but face limitations in speed, balance, and endurance. Level 3 individuals rely on mobility aids for walking and supervision for stair climbing, using wheelchairs for longer distances. Level 4 indicates a lack of self-mobility, with the child only able to sit with support and requiring a wheelchair for transportation. Level 5 denotes complete dependence in all settings, with limitations in maintaining an antigravity posture and mandating wheelchair use for transportation (*Alshryda & Wright, 2014*).

*Eliasson et al. (2006)* introduced the MACS in 2006, a five-level scale tailored for evaluating upper limb function in children aged 4 to 18. This system offers a comprehensive framework for assessing a child's capacity to manipulate objects, accomplish daily activities, and request assistance when needed. At level I, children can manipulate objects with minor limitations in accuracy that do not significantly impact their daily activities. Level II involves slower and reduced quality activities, but the child can find alternative ways to perform tasks without hindering their routines. Level III signifies reduced speed and limited success in hand activities, with some tasks requiring assistance. Level IV indicates a significant effort to perform simple activities, necessitating constant help and specialized equipment. Finally, level V denotes complete dependence on assistance for all activities.

The CFCS uses a five-tier scale to evaluate everyday communication skills. At level I, individuals can comfortably communicate at a normal pace. At level II, communication is slower but effective. Level III indicates effective communication only with familiar partners. Level IV involves inconsistent communication with known individuals, and level

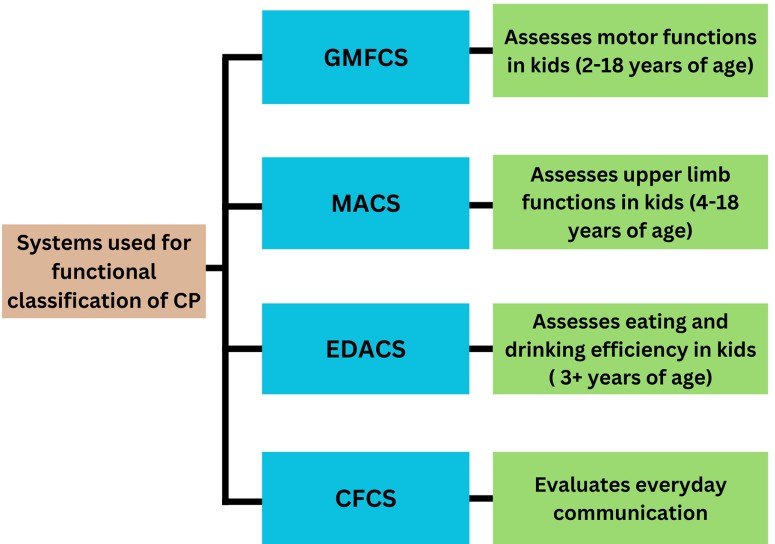

**Figure 1 Systems for functional classification of CP.**

V indicates difficulty communicating effectively and consistently with unknown people (*Paulson & Vargus-Adams, 2017*).

The EDACS is utilized to evaluate the eating and drinking proficiency of children aged three and above. It comprises five levels, with three levels indicating the degree of assistance needed during these activities. Assessing the ability to consume food and beverages offers valuable insights in qualitative analysis. At Level I, individuals can eat and drink safely without assistance, although they may encounter swallowing difficulties with solid foods. Level II signifies safe consumption but at a slower pace, potentially resulting in coughing if eating too quickly. Level III necessitates the consumption of soft and mashed foods. Individuals classified at EDACS levels IV or V are unable to swallow safely and often require tube feeding for nutrition (*Sellers et al., 2014*).

Diagnosis of cerebral palsy is complex and requires multiple assessments, which include neurologic assessment, neuroimaging findings, and recognition of clinical risk factors. The average age when CP can be diagnosed is when the child is 1–2 years old. As a result, various secondary abnormalities increase, and this period can be stressful for parents. Hence, it is imperative to reduce this time. Brain plasticity (the ability of the brain to change its structure and function) in developing infants is more critical. Hence, it is essential to diagnose early so that the treatment and interventions can be started early. Infants with severe brain injuries are the only ones who are diagnosed early using imaging modalities such as Magnetic Resonance Imaging (MRI) and ultrasonography and can benefit from early interventions. General Movements Assessment (GMA) classification is another method to diagnose CP in infants (*Einspieler et al., 1997*). In this method, the clinicians observe the infants for spontaneous movements. Kids without cerebral palsy have fidgety movements. If the movements are repetitive and non-fidgety, the infant has a higher chance of CP. It can be used for newborns from 10–20 weeks. Hammersmith Infant Neurological Examination (HINE) is another neuromotor assessment tool that can be

used for newborns from 2–24 months. It consists of scoring for 26 factors that examine posture, movement, muscle tone, muscle reflexes, and functioning of cranial nerves (*te Velde et al., 2019*).

# CURRENT METHODS OF CP MANAGEMENT AND TREATMENT

Cerebral palsy management focuses on improving the functioning and physical abilities of an individual and managing the secondary complications arising due to CP. Depending on the type of CP, part of the brain compromised, and visible complications, treatment differs for every patient.

Treatment and rehabilitation strategies may encompass a variety of medical interventions, such as acupuncture, orthopedic surgeries, hyperbaric oxygen therapy, physical therapy, speech therapy, occupational therapy, medication management, mobility aids, body weight-supported treadmill training (BWSTT), sensory integration, neurodevelopmental treatment (NDT), hippotherapy, constraint-induced movement therapy (CIMT), and allied therapies tailored to individual needs (*Paul et al., 2022*). Support from parents and surroundings plays a vital role in facilitating proper care and treatment for individuals with Cerebral Palsy and helps increase the quality of life for such patients. Cerebral Palsy management can be done in a better way using a multidisciplinary approach by a team compromising physiotherapist, surgeon, hearing specialist, healthcare social worker, nurse, nutritionist, vocational rehabilitation, pediatric neurologist, gastroenterologist pulmonologist psychiatrist, speech therapist, and special needs teacher. Spasticity is a significant issue in patients with CP and can result in pain, functional loss, and bone and joint deformities. Currently, various approaches are used to control spasticity, which includes medications such as diazepam and baclofen to help relax muscles, physiotherapy exercises, botulinum injections, and surgical management strategies such as lengthening of adductors, hamstrings, and other soft tissues, multilevel foot surgeries, blocking of nerves, joint stabilization, selective dorsal rhizotomy, *etc*. (*Paul et al., 2022*). Children with CP also require help with balancing and dealing with movement disorders to be able to perform maximum day-to-day activities. This is achieved by giving regular physiotherapies, occupational therapies, acupuncture, *etc*., to the patients. Core stability exercises, virtual reality, and whole-body vibrations effectively improve balance. Hand dysfunctioning is another secondary issue associated with cerebral palsy, which can affect single or both hands. In such cases, children face difficulties in moving their hands. One of the techniques used to manage this condition is constraint-induced movement therapy (CIMT). This technique works to improve the hand movement of the affected hand by making use of neuroplasticity. The patient's affected hand is intensively used instead of the unaffected hand to enhance its functioning. Hip-related disorders such as hip dislocation and other related problems affect around 36% of total kids suffering from cerebral palsy. These problems are mainly managed surgically. Orthotic devices, such as ankle foot orthoses (AFOs), are used to manage foot-related problems such as abnormal gait, muscle weaknesses, deformities, *etc.* Various robot-assisted devices and wearables are used for upper-limb and lower-limb rehabilitation

(*Paul et al., 2022*). Powered mobility devices that support standing upright and walking enable the movements of legs similar to that of normal walking, inducing flexion and extension of the hip, knee, and ankles. This positively affects the strength, tone, and functioning of muscles, bowel functions, endurance, flexibility, and overall fitness (*Schmidt-Lucke et al., 2019*). These devices support CP children in standing, walking, or performing movements and reduce the energy expenditure of the child, reducing fatigue. Dynamic standing helps to increase the passive range of motion and reduces spasticity in the hip among non-ambulatory children with cerebral palsy (*Tornberg & Lauruschkus, 2020*). These technologically advanced devices are advantageous due to their ability to operate for extended periods, maintain a consistent speed, and execute patterns and repetitions (*Paul et al., 2022*).

Epilepsy is another common secondary outcome of cerebral palsy, resulting in children having seizures. It is managed by the use of drugs such as valproic acid, vigabatrin, clonazepam, clobazam, levetiracetam, *etc.*, to name a few. Children with cerebral palsy often have behavioral issues such as ADHD (attention-deficit/hyperactivity disorder), anxiety, depression, *etc.* Experienced psychologists play a vital role in helping CP children and parents strategically manage these issues. Due to muscle weakness in cerebral palsy, patients often suffer from swallowing disorders, drooling, and speech-related problems. Modification of food, careful feeding techniques, tongue control, medications, surgery (ligation of duct, removal of salivary glands), and speech therapy are the approaches that can be used to manage these conditions. Muscle weakness and inadequate posture control also may lead to respiratory problems caused by the entry of food particles into the respiratory tract. This may lead to the growth of bacteria in the respiratory system, causing respiratory failure and death. Lifestyle modifications, neck control exercises, respiratory hygiene, *etc.*, are used to manage this condition.

Sleep disorders are very commonly seen in CP patients and are one of the causes of behavioral disorders. It affects the quality of life of patients as well as parents, producing a psychological burden on them. Polysomnography (sleep test) is used to evaluate sleep-related issues and can be treated by surgical interventions or sensory system stimulation.

## Drawbacks of current methods of CP patient care

CP is a complex neurological disorder that requires specialized, multidisciplinary care. Intensive rehabilitation programs for CP are run in several countries, but it is challenging to obtain reliable data regarding the effectiveness of the practices. While current methods of CP patient care have significantly improved over the years, several drawbacks remain. A few are listed below:

### Fragmented care

Care for CP patients often involves multiple specialists, including neurologists, orthopedic surgeons, physical therapists, speech therapists, and occupational therapists. This multidisciplinary approach, while necessary, can lead to fragmented care due to poor coordination between providers. Inconsistent communication between healthcare providers can result in duplicated efforts, conflicting treatment plans, and important

information being overlooked. This may affect the overall treatment outcomes for the patients as each specialist might focus on their domain of expertise instead of focusing on the holistic approach to improve the patient's quality of life. It may add additional financial and operational burdens for parents or caregivers as they would require more money to pay specialists from different fields and manage the appointment schedules, leading to stress and affecting the patient's quality of care.

### Accessibility and availability

One of the significant drawbacks in the management of CP is the limited accessibility to quality and specialized healthcare services. Various programs such as HINE and GMA evaluate the patient's condition, but their implementation is limited due to inadequate training and certifications for the providers (*Hornby et al., 2024*). Access to specialized care can be limited in rural areas due to a lack of infrastructure and trained specialists, leading to disparities in the quality of care patients receive based on their location. Resource scarcity often results in delays in diagnosis and treatment, hampering patients' quality of life. Patients usually face long wait times for specialist appointments, delaying essential treatment and intervention. Additionally, families may skip the diagnosis and treatments due to the higher associated costs. Access to physiotherapy or specialized therapy for neurodevelopment in CP kids can be limited due to various social, geographical, economic, and availability factors. Transitioning from pediatric to adult care for CP patients might be challenging as fewer healthcare providers are willing to manage the complex needs of the adults. The lack of continuity in patient care eventually results in complications over time. Various programs such as HINE and GMA evaluate the patient's condition, but their implementation is limited due to inadequate training and certifications for the providers.

### Financial burden

Neurological disorder treatment often comes with a high price tag. The cost of diagnosis and treatment of CP are directly related to the severity of the disorder. Severe forms involve more hospital admissions, leading to respiratory disorders and increasing the cost of treatment (*Ismail et al., 2022*). The medical expenses of a family with a CP child include the cost of surgeries, medications, different types of therapies, drugs, *etc*. Additionally, there is also a need for certain modified or assistive devices such as mobility devices, communication devices, *etc*. These costs are difficult to manage for some families and can be prohibitive for many. Also, it is observed that insurance may not cover all necessary treatments or not support therapy session expenses. Families might have to modify the house and furniture according to the needs of CP patients. Many parents have to compromise with their jobs and working hours, resulting in the loss of jobs and opportunities, lower incomes, and career setbacks. These factors, at times, lead to poor mental health of caregivers, leading to depression.

### Technological and resource constraints

Advanced technology has contributed significantly to the rehabilitation field. These equipment are costly, and hence, some care facilities may lack access to the devices, hindering the effectiveness of treatments. High-tech devices require timely maintenance

and updates, which can be logistically challenging and expensive. Additionally, the lack of trained professionals to operate these devices results in ineffective use and reduced patient benefits. Some patients with cerebral palsy require specialized, technologically advanced solutions such as communication devices, powered mobility devices, and custom-made orthotics, which are highly expensive and unaffordable for middle to lower-class income populations. To start early treatments and interventions and reduce the risk of secondary abnormalities, early diagnosis of CP is required. Imaging modalities used for these are not sensitive enough to recognize the brain injuries in premature infants and may lead to delays in diagnosis. Other methods include GMA assessment, which requires experts to put in the time and effort to get training for the assessment. Also, it is important that doctors be trained to use advanced AI-ML-based systems that can significantly help provide quantitative feedback in CP diagnosis and treatment. Limited resources can restrict the availability of innovative therapies and interventions that could improve patient outcomes.

### Inadequate personalized care

Children with CP have a wide range of problems and conditions (neurological and muscular); however, the interventions are more generalized and do not address the specific impairments. Current methods for cerebral palsy diagnosis and treatment use various techniques and approaches, but one significant drawback is the lack of a targeted approach. This One-Size-Fits-All approach is the major hurdle seen in the treatment of CP patients. Treatment plans are sometimes not sufficiently individualized to cater to the specific needs of each patient, leading to suboptimal outcomes.

### Limitations of using drugs

There are several limitations to the use of drugs for CP patients due to the potential side effects and the variability in the outcomes. Side effects such as drowsiness, fatigue, and issues related to the gastrointestinal tract due to drugs like muscle relaxants, antispastic agents, *etc.*, impact the overall quality of life of the patients. The process of finding the right drug and dose often involves trial and error, causing delays in treatment and stress to the families. The treatment of spasticity in cerebral palsy involves the use of medications to relax muscles, but these medications can come with potential side effects. For extreme cases of spasticity, intrathecal baclofen is administered through implantable pumps, although this treatment is costly and provides relief for a shorter duration (*Paul et al., 2022*). The diverse approaches to treatment can lead to inconsistent outcomes, posing a challenge for standardizing patient care.

### Public awareness

Cerebral palsy can be devastating, and patients have to face a lifetime of challenges. There is very little awareness about CP, its causes, symptoms, and the mental and physical abilities of the patients. This lack of awareness is also due to the complexity of the condition. Many people have certain misconceptions about CP, such as it is a contagious disease or that all patients with CP have compromised cognitive abilities. Awareness is crucial for early diagnosis and management plans. There is often a lack of awareness and

understanding of CP among the general public and even within the medical community, which can impact early diagnosis and intervention.

### Research and development barriers (use of engineering technology in medicine)

More research is needed to focus on the long-term outcomes of various treatments and interventions for CP. Challenges in translating research findings into practical, widely available treatments and therapies persist, slowing down progress in CP care. It is challenging to find and recruit participants for the research due to the unique needs of each patient. CP research requires a multidisciplinary approach and can be expensive and administratively challenging. Complex ethical considerations for CP research to ensure patient safety can slow down the research.

Addressing these drawbacks requires a concerted effort to improve coordination of care, increase accessibility, reduce financial burdens, provide holistic and personalized care, invest in technological advancements, enhance professional training, and foster ongoing research and innovation.

## Prior research

Multiple authors have discussed the current methods of diagnosis and treatment that are in clinical use for cerebral palsy. The study by *Herskind, Greisen & Nielsen (2015)* emphasizes the benefits and need for early diagnosis and proposes a combination approach using neuroimaging and GMA for early detection. However, there is no substantial evidence supporting the approach.

Another study (*Hadders-Algra, 2014*) reviews challenges related to early diagnosis and interventions for cerebral palsy. The article discusses predictive techniques, including neuromotor exams, neuroimaging, and neurophysiological assessments. Although combining these techniques shows promise in early interventions, more research is required to establish evidence.

*Novak et al. (2017)* conducted a comprehensive review examining the latest evidence for early and accurate diagnosis of cerebral palsy. Traditionally, diagnosis occurred between 12 and 24 months of age, but recent advancements allow diagnosis before 6 months of corrected age. The study underscores early intervention post-diagnosis's significance in maximizing neuroplasticity and functional outcomes. Through a summary of evidence on early interventions specific to cerebral palsy, the authors outline strategies aimed at improving outcomes for children affected by the condition.

While all the above studies focus on the use of GMA and other conventional methods for cerebral palsy early diagnosis and interventions, the *Zhang (2017)* study tells the importance of multivariate analysis and machine learning in enhancing CP diagnosis, treatment, and patient care. The review offers valuable insights into the use of multivariate analysis (MVA) and machine learning (ML) in CP research. However, it lacks a detailed analysis of the required datasets, implemented algorithms, features extracted, and comparative analysis of all the machine learning and deep learning algorithms used for CP patient care.

**Table 1 Prior research: scope, observation and limitations of the selected literature.**

| Ref | Year | Objective | Merits | Demerits |
|---|---|---|---|---|
| Herskind, Greisen & Nielsen (2015) | 2014 | Importance and need for early identification and interventions for cerebral palsy was discussed. | Detailed explanation of need for early identification and intervention for CP. | The review does not cover the implications of using AI for early identification of CP or other aspects of CP patient care such as classification and treatment. |
| Hadders-Algra (2014) | 2014 | This article examines the opportunities and challenges associated with the early diagnosis and intervention of cerebral palsy. | The review discusses brain lesions, their occurrence, and the differences in diagnosis and interventions based on the timing and location of the brain lesion. | Review does not cover the use of AI for early identification and intervention for CP. |
| Novak et al. (2017) | 2018 | Review the most reliable evidence to identify cerebral palsy early and accurately, and summarize the most reliable evidence on early interventions specific to cerebral palsy. | Detailed overview of the different diagnosis and treatment methods for CP. | Review does not cover the use of AI for early identification and intervention for CP. |
| Zhang (2017) | 2015 | To explore and summarize the utilization of multivariate and machine learning approaches in cerebral palsy research through the identification of relevant multivariate studies. | The article reviews the articles using MVA and ML approaches for CP research. | The review uses the studies only related to pediatric patients. It does not cover the quantitative analysis of CP research. |

**Note:**
CP, cerebral palsy; MVA, multi variate analysis; ML, machine learning.

Our systematic literature review provides a comprehensive overview of the recent advancements, trends, and challenges pertaining to the application of machine learning and deep learning methodologies in the early prediction, diagnosis, classification, treatment, and interventions for cerebral palsy. Additionally, our review conducts a comparative analysis of the utilized techniques and identifies research gaps, thereby offering insights into potential avenues for future research. The primary objective of this article is to critically examine the existing literature concerning the implementation of machine learning and deep learning techniques in the care of cerebral palsy patients, address the identified research gaps, and propose an evidence-based framework that can be translated into clinical practice for accurate and timely diagnosis and interventions in cerebral palsy management.

Table 1 highlights some of the literature review studies carried out in the domain of the use of AI in CP patient care.

## Motivation

This extensive review provides an overview of the recent developments in the prediction, diagnosis, and treatment of cerebral palsy through machine learning and deep learning methodologies. By critically analyzing existing research studies and their findings, this article aims to address key research questions in the field. Furthermore, this article provides a systematic literature review (SLR) that, to the best of the author's knowledge, represents one of the first comprehensive analyses of AI-based methods for the early

**Table 2 Research goals.**

| Number | Research questions | Importance |
|---|---|---|
| RQ1 | What is the distribution of published articles concerning the application of AI in CP patient care, categorized by year, subject area, country and publication type? | It assists in identifying the timing, location, and entity responsible for conducting research on the subject matter. |
| RQ2 | What datasets are accessible for various categories of issues in CP patient care? | It assists in finding a dataset with appropriate information for good research outcomes. |
| RQ3 | What are the primary artificial intelligence techniques utilized for early prediction, diagnosis, and treatment of cerebral palsy (CP)? Additionally, what evaluation criteria are commonly employed in assessing these methods? | It supports the identification of suitable artificial intelligence methods for the early prediction, diagnosis, and treatment of today's CP. It assists in choosing the right assessment criteria for measuring performance. |
| RQ4 | What are the primary obstacles and issues encountered by current AI based approaches to early prediction, diagnosis, and treatment of CP? | It helps in examining key challenges in researching AI methods for the early prediction, diagnosis, and treatment of cerebral palsy, while also highlighting the advantages and drawbacks of existing studies and solutions. |
| RQ5 | What are the future directions for developing a strong and dependable AI-based system for early prediction, diagnosis, and treatment of CP? | It helps in delving into crucial research areas that remain unexplored. |

**Note:**
  RQ 1, research question 1; RQ 2, research question 2; RQ 3, research question 3; RQ 4, research question 4; RQ 5, research question 5.

prediction, diagnosis, and treatment of cerebral palsy. The review emphasizes the importance of delivering precise and effective care for individuals with this condition.

## Research goals

The objective of this review is to assess the present state of early prediction of cerebral palsy in infants, identify the diverse abnormalities linked with cerebral palsy, classify the condition, and evaluate existing treatment and rehabilitation approaches. It scrutinizes multiple studies showcasing the utilization of machine learning to enhance diagnostic and therapeutic outcomes while pinpointing research voids to construct a machine learning/ deep learning framework suitable for clinical implementation, facilitating accurate diagnosis and effective rehabilitation. Table 2 summarizes the research inquiries explored in this literature review.

## Contributions of the study

Here are the primary contributions of our literature review:

- A comprehensive review of research investigations identified using the Preferred Reporting Items for Systematic Reviews and Meta-Analyses (PRISMA) methodology, focusing on the early anticipation, diagnosis, categorization, and management of cerebral palsy, incorporating AI techniques such as machine learning and deep learning.
- An in-depth analysis of the volume and reliability of standardized datasets.
- Discussions on classification methods, their practical implications, and challenges and issues in CP patient care using AI.
- Exploration of various evaluation metrics used in CP early prediction, diagnosis, classification, and treatment.

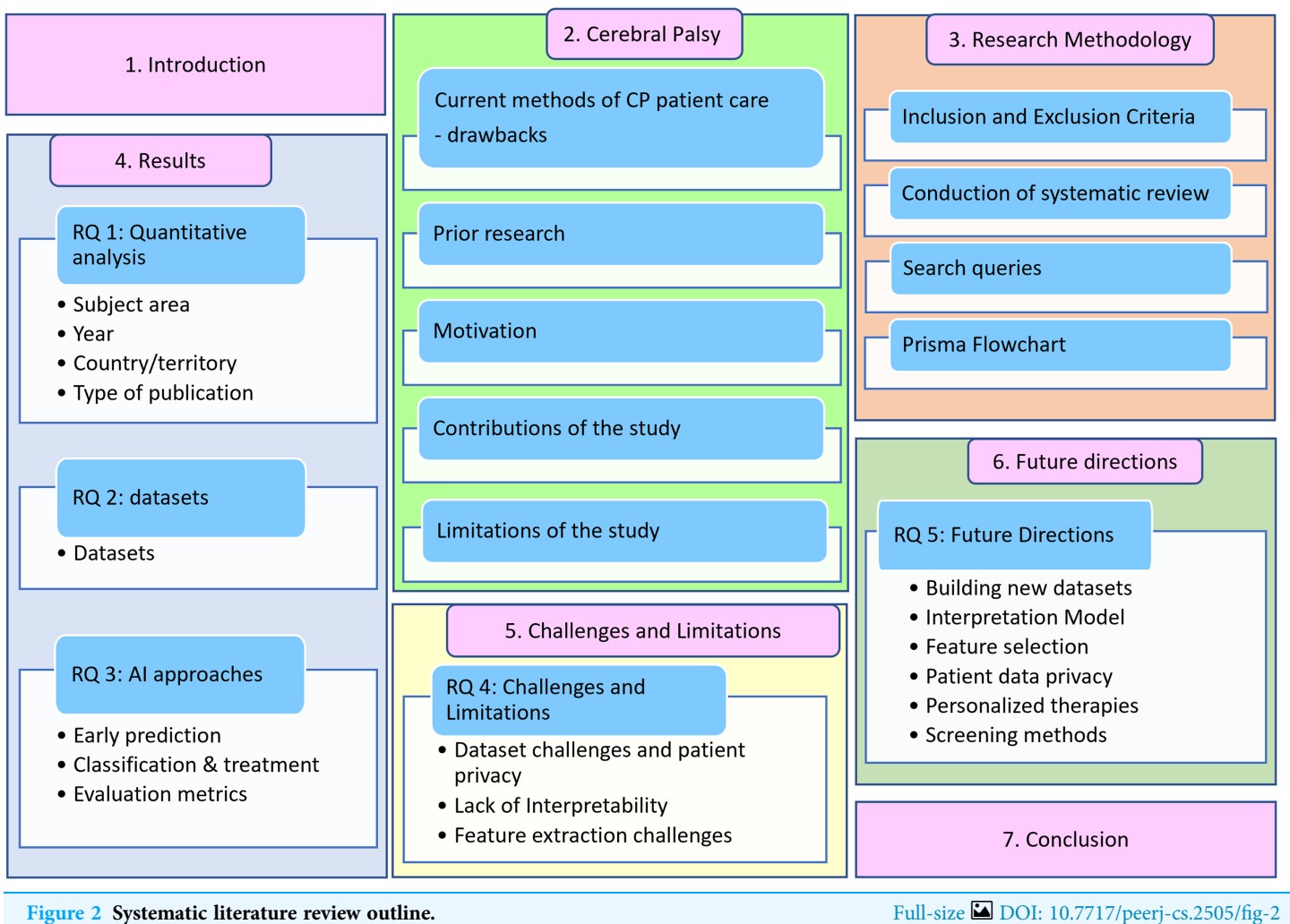

**Figure 2 Systematic literature review outline.**

• Formulating future research directions and providing insights to guide researchers in identifying the most dependable and precise diagnosis and treatment approaches for cerebral palsy.

## Limitations of the study

The review does not explain the complete implementation of AI for CP patient care. Still, it uses some research articles to give an overview of the AI methods implemented for CP early prediction, classification, and treatment. Certain relevant research studies may have been excluded from our review due to limitations within the scientific database, the specific keywords used during the search, and the duration of the review. From 2014 to 2024, the author selected only 69 studies. The manual screening of studies obtained from library databases such as SCOPUS and Web of Science (WoS) was carried out with assurance. Articles from open-access articles such as Peerj and MDPI are added to the study. The review is based on the application of AI in CP patient care, which includes various

subtopics such as implementation of AI for early diagnosis of CP, implementation of AI for diagnosing secondary abnormalities associated with CP, Classification of patients based on the severity of condition, CP management, *etc.* Hence, the methodologies, datasets, evaluation metrics, *etc.*, may have heterogeneity, making it challenging to compare the results and draw a unified conclusion. Although the review aims to enable the practical implementation of AI in CP patient care, certain ethical issues may limit the clinical implementation.

The article is organized as follows: "Current Methods off CP Management and Treatment" discusses the current methods of CP patient care, prior research, research goals, contributions, and limitations. "Survey Methodology" introduces the proposed methodology, explaining the exclusion and inclusion criteria, systematic literature review, and search queries. "Results" discusses the findings and answers to research questions RQ1, RQ2, and RQ3. "RQ 4—Challenges and Limitations of using AI for CP Patient Care" examines the challenges and constraints associated with implementing AI in cerebral palsy patient care. "RQ 5—Future Directions" delves into future directions, followed by conclusions for the review article. The structure of this review is illustrated in Fig. 2.

## SURVEY METHODOLOGY

### Inclusion and exclusion criteria

The authors established a set of standards for selecting and rejecting research articles, as outlined in Table 3, to identify the most suitable research articles for the review.

The screening procedure followed the following process to establish the inclusion and exclusion criteria.

(i) Initial abstract screening: Evaluate research abstracts to eliminate irrelevant articles by assessing their alignment with relevant knowledge and keywords. Consider abstracts that meet at least 40% of the inclusion criteria for further consideration.

(ii) Full-text evaluation: Excluded articles that do not correspond to or contribute to the search query outlined in Table 4. This involves disregarding articles with abstracts that only partially address the search query.

(iii) Quality appraisal: Subject the remaining research articles to a quality assessment, excluding any that do not meet the specified criteria.

### Conduction of systematic review

The following steps were utilized to choose the appropriate articles for this review:

Scopus and WoS were used to search the relevant literature articles for the review. Table 4 highlights the search queries used in SCOPUS and Web of Science. Search query used for Scopus was '("Cerebral Palsy") AND ("Machine Learning" OR "Deep learning" OR "Reinforcement learning" OR "Artificial Intelligence" OR "Neural Network") AND ("Diagnosis" OR "Treatment" OR "Rehabilitation" OR "Prognosis" OR "Training" OR "Classification").' There were 293 articles between 2014–2024. Further filters for language were applied, which gave 290 results. Out of these, only research articles and review articles were selected, which accounted for 191 results. A similar process was followed to find the research articles through the Web of Science database, which yielded 160 results. The next

**Table 3 Inclusion and exclusion criteria summary.**

**Inclusion criteria**

Articles should be original research articles.

Research articles that were released between 2014–2024

Research article that answers at least 1 research question.

Keywords should be included in the abstracts, titles or full text of articles.

**Exclusion criteria**

Articles which are now written in the English language.

Duplicate research articles

Research articles whose' full text is not accessible.

Research articles that are irrelevant to use of artificial intelligence in cerebral palsy patient care.

**Table 4 Search queries used for SCOPUS and Web of Science.**

| Database | Query | Initial results |
|---|---|---|
| Scopus | ("Cerebral Palsy") AND ("Machine Learning" OR "Deep learning" OR "Reinforcement learning" OR "Artificial Intelligence" OR "Neural Network") AND ("Diagnosis" OR "Treatment" OR "Rehabilitation" OR "Prognosis" OR "Training" OR "Classification") | 351 |
| Web of Science | ("Cerebral Palsy") AND ("Machine Learning" OR "Deep learning" OR "Reinforcement learning" OR "Artificial Intelligence" OR "Neural Network") AND ("Diagnosis" OR "Treatment" OR "Rehabilitation" OR "Prognosis" OR "Training" OR "Classification") | 212 |

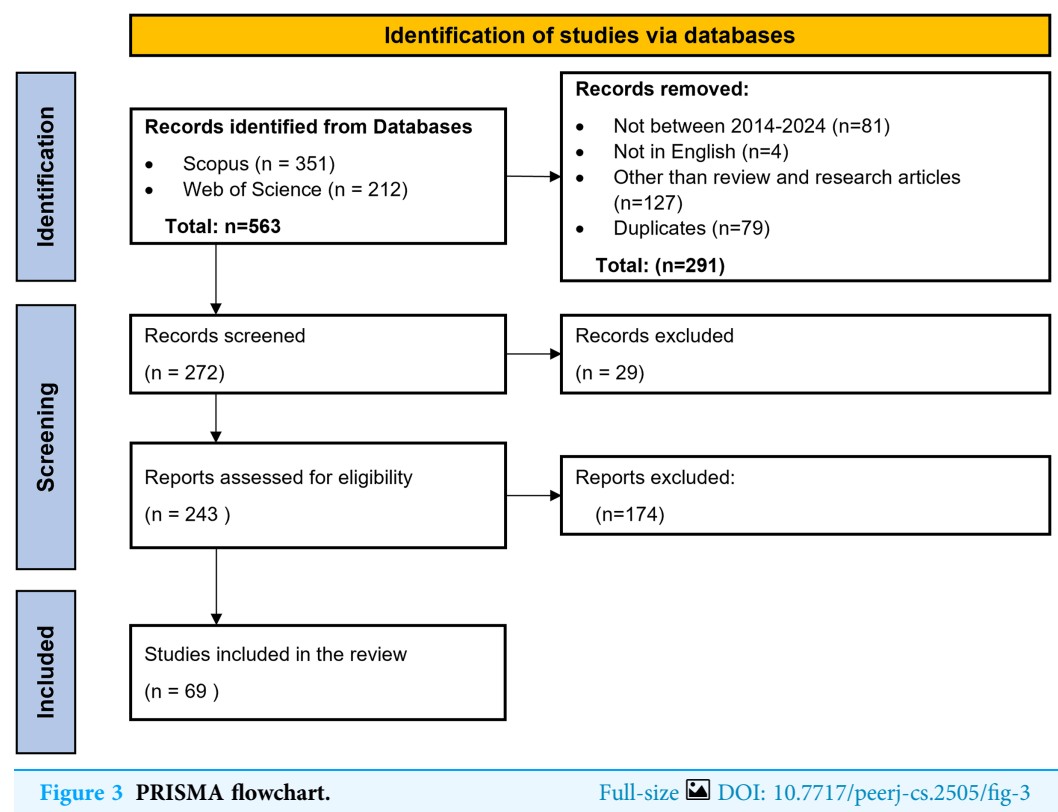

**Figure 3 PRISMA flowchart.**     

step included removing the duplicate articles from both databases, which was done by reading the titles of the articles. A total of 79 duplicate articles were removed. The subsequent screening phase entailed applying the inclusion and exclusion criteria. This entailed reviewing the titles and abstracts of the articles. Subsequently, following a thorough analysis of the full texts and application of the inclusion and exclusion criteria, a total of 69 articles were selected. These articles were then critically studied for this review to find the research gaps, limitations, and directions for further research in the field of AI application for CP patient care. Figure 3 represents the PRISMA flowchart for selecting articles for conducting the review.

## RESULTS

This section provides a summary of the outcomes of the literature analysis, addressing the research questions derived from the review of 69 articles. "Inclusion and exclusion criteria" delves into Research Question 1 (RQ1), focusing on quantitative analysis of the topic. "Conduction of systematic review" addresses Research Question 2 (RQ2), which examines the datasets utilized in studies within the field of interest. RQ3 discusses various machine learning and deep learning techniques for cerebral palsy patient care. It also gives information about the evaluation parameters used for these research studies. RQ 4 is addressed in "Results", where the challenges and limitations to the implementation of AI in Cerebral Palsy patient care are discussed. "RQ 4—Challenges and limitations of using AI for CP patient care" addresses RQ 5, which includes information about future research directions in the AI application field for CP patient care.

### RQ1—Quantitative analysis of literature on the use of AI ML for CP patient care

A total of 162 search results through SCOPUS and 160 through Web of Science were found for the articles related to the implementation of AI in CP patient care after applying filters for language, publication year, and article type.

Figure 4 shows the quantitative analysis of literature obtained out of the 162 articles obtained through SCOPUS. The highly interdisciplinary aspect of this area can be seen in the percentage distribution chart in Fig. 4A. A total of 31.7% of research articles were from medical journals, 13.8% from engineering journals, 13.2% from computer science journals, and 12.3% from neuroscience journals. Together, these contributed to 71% of the total publications. The rest of the contributors included biochemistry, health professions, chemistry, multidisciplinary, psychology, and materials science. This data is essential to understand the wider scope and integration of the medical and engineering field required to treat a complex neurological disorder like cerebral palsy. Figure 4B shows the distribution by type of the document. A total of 26 articles were review articles, and 136 were journal articles. Of the total articles published from 2014 to 2024, only one was published in 2014, whereas in 2023, 42 articles were published. Nineteen articles were published in 2024 until March. Figure 4C shows the increasing trend in publications year-wise. Figure 4D shows the country-wise research contributions. The awareness (as well as advanced healthcare facilities) is more in the developed countries about prenatal and

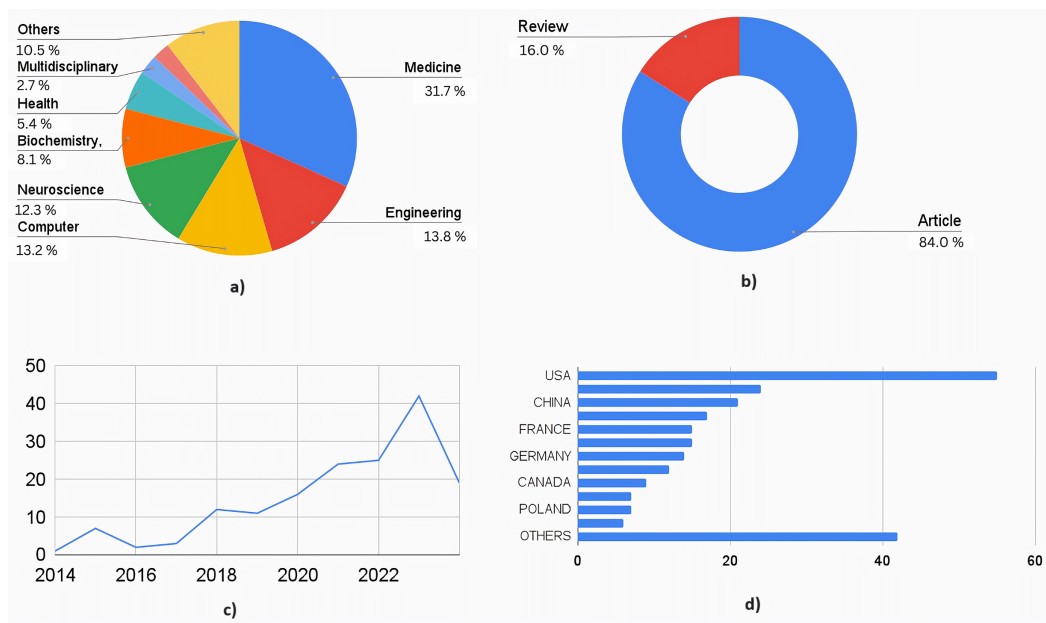

**Figure 4  Quantitative analysis of literature based on database from Scopus.** (A) Documents by subject area. (B) Documents by type. (C) Documents by year. (D) Documents by countries/territories.

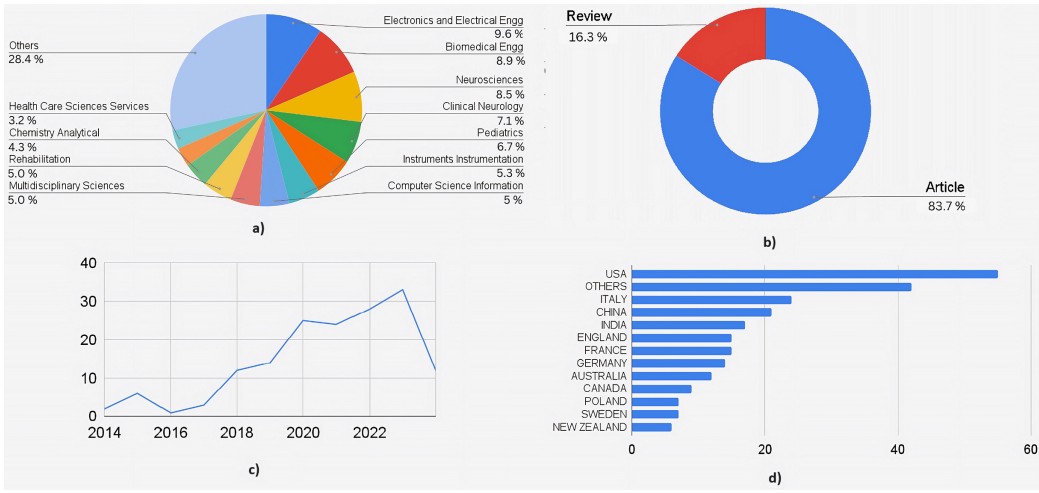

**Figure 5  Quantitative analysis of literature based on database from Web of Science.** (A) Documents by subject area. (B) Documents by type. (C) Documents by year. (D) Documents by countries/territories.

postnatal interventional therapies used to detect and treat neurological conditions in fetal and newborn babies, which makes it evident that developed countries contribute to 70% of total publications. The US contributed the most articles, followed by Italy and the United Kingdom.

Figure 5 shows a similar analysis but from the literature indexed by the Web of Sciences (WoS). A total of 160 articles were selected for the quantitative analysis. Figure 5A shows the distribution based on subject area: 18% belonged to engineering applications in

healthcare, followed by medical streams of neuroscience, clinical neurology, and pediatrics, which contributed to another 24%. Computer engineering, Instrumentation engineering, multidisciplinary sciences, and rehabilitation each accounted for an average of 5%, covering another 20% of the documents. Figure 5B shows documents by submission type. One hundred thirty-four were research articles, 26 were review articles, and the remaining were from conferences. Figure 5C shows the graph of the publications from 2014 to 2024. A similar graph to that of documents from Scopus can be seen (where a dip in the number of research articles is seen from 2015 to 2016). Figure 5D shows the contributions by countries where the USA has contributed to the maximum number of articles, followed by Italy, China, and combined European Countries. The analysis shows that there is enormous scope for using AI ML not just in diagnosis but also in prognosis and management of cerebral palsy.

## RQ 2—Datasets

Real-world datasets are often complex, irregular, messy, and unstructured. Achieving a suitable equilibrium among quantity, relevance, and data quality is crucial. Datasets determine the performance, accuracy, and reliability of the models. Few published datasets have been used for the research in the domain of early prediction of CP using AI. Table 5 summarizes these datasets used in the literature.

Following are the few commonly used datasets for CP early prediction.

- **Moving Infants in Red Green Blue Depth (MINI RGBD)**—MINI RGBD stands for moving infants in RGB-D. This is the dataset comprising videos of 12 infants lying in the supine position. The age of infants in this dataset is around 0.6 years. Each video contained 100 frames with a color video resolution of 640 * 480. A skinned multi-infant linear body model (SMIL) was used to form this dataset to ensure anonymity in the patient's identity. Experts labeled the dataset to distinguish babies with and without fidgety movements based on GMA. MINI RGBD is the most popular dataset used for research focusing on cerebral palsy early prediction (*Hesse et al., 2019*; *McCay et al., 2022*; *Wu et al., 2023*; *Sakkos et al., 2021*; *Devarajan & Khader, 2023*).
- **Royal Victoria Infirmary (RVI) 38 Dataset**—The RVI 38 Dataset consists of 38 authentic videos depicting infants aged 3–5 months post-term, recorded during regular clinical activities at the Royal Victoria Infirmary (RVI) in Newcastle upon Tyne. These videos were filmed using a handheld SONY DSC-RX100 advanced Compact Premium Camera with a resolution of 1,920 * 1,080. On average, the videos in this dataset have a length of 3 min and 36 s (*McCay et al., 2022*; *Wu et al., 2023*).
- **Royal Victoria Infirmary (RVI) 25 Dataset**—The RVI 25 Dataset consists of 25 videos of infants in the supine position, recorded during standard clinical procedures at the Royal Victoria Infirmary (RVI) in Newcastle upon Tyne. These videos vary in length from 1 to 5 min (*Sakkos et al., 2021*).
- **MODYS-video**—The RVI 38 Dataset consists of 38 authentic videos depicting infants aged 3–5 months post-term, recorded during regular clinical activities at the Royal Victoria Infirmary (RVI) in Newcastle upon Tyne. These videos were filmed using a

**Table 5 Some common datasets used for research on CP using AI.**

| Dataset | Year | Subjects | Age group | Created by | Used in literature | Links |
|---|---|---|---|---|---|---|
| MINI RGBD | 2018 | 12 | 0–7 months | Hesse, Nikolas, Christoph Bodensteiner, Michael Arens, Ulrich G. Hofmann, Raphael Weinberger, and A. Sebastian Schroeder. | *McCay et al. (2022)*, *Wu et al. (2023)*, *Sakkos et al. (2021)*, *Devarajan & Khader (2023)*, *Mathis et al. (2018)* | https://www.iosb.fraunhofer.de/en/competences/image-exploitation/object-recognition/sensor-networks/motion-analysis.html |
| RVI 38 | 2022 | 38 | 3–5 months | Royal Victoria Infirmary (RVI) in Newcastle upon Tyne, UK. | *McCay et al. (2022)*, *Wu et al. (2023)* | Available upon request—Edmond S. L. Ho (Shu-Lim.Ho@glasgow.ac.uk). |
| BabyPose, (*Migliorelli et al., 2020*) | 2020 | 16 | Preterm infants | Lucia Migliorelli, a, * Sara Moccia,a,b Rocco Pietrini,a Virgilio Paolo Carnielli,c and Emanuele Frontonia | *Devarajan & Khader (2023)* | https://zenodo.org/records/3891404 |
| MIA (*VRAI, 2024*—Vision) | 2018 | 1 | Pre term (37+1 weeks of gestational age) | VRAI—Vision, Robotics and Artificial Intelligence Department of Information Engineering Universit`a Politecnica delle Marche *Via* Brecce Bianche 12, 60131 Ancona, Italy | *Devarajan & Khader (2023)* | https://vrai.dii.univpm.it/mia-dataset |
| MODYS-video | 2021 | 34 | Mean age—14 years 2 months | Haberfehlner, H., Bonouvrié, L. A., Stolk, K. L., van der Ven, S. S., Aleo, I., van der Burg, S. A., van der Krogt, M. M., & Buizer, A. I. | *van der Krogt & Haberfehlner (2021)* | https://zenodo.org/records/5638470 |

**Note:**
MINI-RGBD—moving infants in red green blue depth; RVI 38, royal victoria infirmary 38; MIA, motion infant analysis.

handheld SONY DSC-RX100 advanced Compact Premium Camera with a resolution of 1,920 * 1,080. On average, the videos in this dataset have a length of 3 min and 36 s (*McCay et al., 2022*; *Wu et al., 2023*). These coordinates are accompanied by clinical scores from the Dyskinesia Impairment Scale (DIS). The recordings were conducted during the "lying in rest" and "sitting in rest" sessions as part of the DIS assessment at three different time intervals within a clinical trial investigating the impact of intrathecal baclofen. The participants, with an average age of 14 years and 2 months (standard deviation 4.0), included 26 males. The range of their gross motor function classification system level spanned from IV to V, while their manual ability classification system level varied from III to V. The original videos are 4–35 s long with a resolution of 720 × 575 pixels and are sampled at 25 Hz. Stick figures were added to complement the data for context and ease of understanding. These figures were created from the 2D coordinates extracted with a likelihood >0.8. Clinical scoring was conducted by three trained experts (according to the DIS) on the original videos. Within the "lying in rest" and "sitting in rest" activities, the amplitude and duration of dystonia and choreoathetosis of the trunk, proximal right arm, proximal left arm, proximal right leg, and proximal left leg were scored on a 0–4 ordinal scale and calculated towards a percentage score between 0–1. The dataset offers the potential for employing a machine learning approach to automatically evaluate dystonia and choreoathetosis in children with dyskinetic cerebral

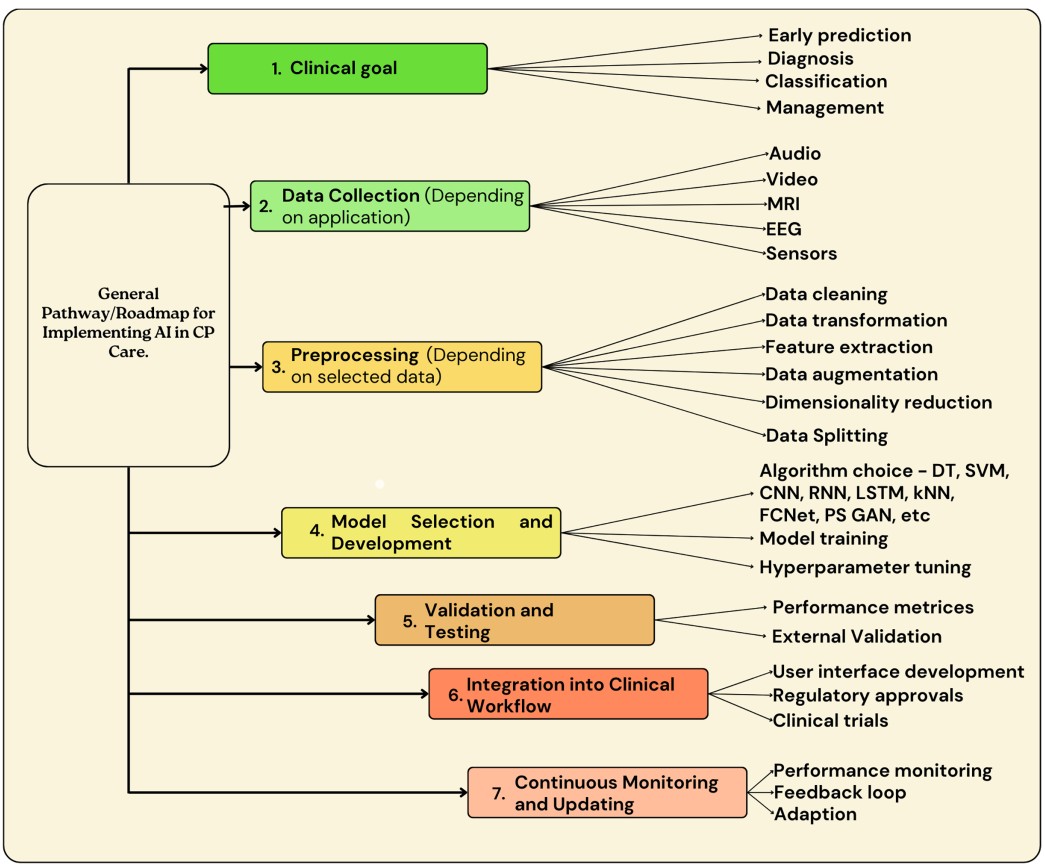

**Figure 6 Overview of generalized roadmap that can be used for implementation of AI in CP patient care.**

palsy. This is achieved through the utilization of 2D coordinates extracted from video recordings of body points (*van der Krogt & Haberfehlner, 2021*; *Mathis et al., 2018*; *Haberfehlner et al., 2023*).

## RQ 3—What are the primary artificial intelligence techniques used for early prediction, diagnosis, and treatment of cerebral palsy?

Artificial intelligence can be used in different areas of cerebral palsy care.

Figure 6 highlights the generalized roadmap that can be used to implement AI in CP patient care.

As shown in Fig. 6, the main steps for AI implementation in CP patient care involve goal identification, where the purpose or goal for AI implementation should be identified. The next step includes data collection, followed by data pre-processing. Figure 7 highlights some of the preprocessing techniques used in previous research. Once the data is pre-processed, the step involves AI model building and implementation, followed by validations, testing, integration into clinical workflow, continuous monitoring, and updating.

- Removal of artifacts like eye movements and muscle activity.
- Identification of bad channels.
- Data re-referencing.
- Segmentation of EEG into 2 second epochs.

**EEG data**

- Data conversion – DICOM to NIfTI format.
- Data standardization by z-score.
- Linear registration to align data to MNI152 standard T1 structure.

**MRI data**

- Splitting dataset into learning and test datasets.
- Remapping anomalous joint positions.
- Removal of outlier joint positions.
- Data normalization.
- Median filtering to eliminate outliers.
- Interpolation to address missing data.
- Video screening and pixel tracking.
- Cropping to ensure only infants were visible.

**Video data** 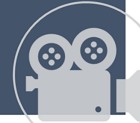

- Converting of image to greyscale.
- Image resizing to enhance pixel brightness.
- Image cropping.
- Image segmentation.
- Image resampling.
- Local contrast normalization.

**Image data** 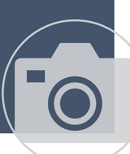

**Figure 7** Some of the preprocessing techniques used in previous studies on implementation of AI for CP care.

Table 6 highlights the basic advantages and disadvantages of some of the basic AI algorithms used in CP patient care research.

Not much research has been conducted into the application of artificial intelligence to cerebral palsy patient care, but there are few use cases. One such is a mobile application called Voiceitt. Voiceitt can be used by people with speech impairments suffering from conditions such as stroke, Cerebral palsy, Parkinson's, and Down's syndrome to communicate. This application uses machine learning to recognize the speakers' unique speech patterns and mispronunciations and create output audio or text by normalizing the speech for everyone to understand (*PBS, 2019*).

Another application of machine learning in Cerebral Palsy patient care is research being carried out at the Massachusetts Institute of Technology (MIT), where the researchers are working on the development of a technology for the evaluation of CP patients remotely.

**Table 6 Basic advantages and disadvantages of some machine learning and deep learning models used in CP patient care research.**

| Algorithms | Advantages | Disadvantages |
|---|---|---|
| Regression analysis | • Simple modelling.<br>• Strong interpretability.<br>• Effective with smaller datasets.<br>• Less resource intensive. | • Difficulty with non-linear problems.<br>• Difficulty with multi collinearity problems.<br>• Overfitting risk.<br>• Weaker performance than ensemble learning and regression analysis.<br>• Sensitive to unbalanced data.<br>• Dependent on quality of data.<br>• Higher sensitivity to outliers. |
| GCN | • Higher accuracy than ML method (CIMA).<br>• Complex data handling.<br>• Higher accuracy in CP detection and classification.<br>• Can integrate with multimodal data.Scalable. | • No significant accuracy as compared to GMA tool.<br>• High complexity.<br>• Requires large amount of high-quality data.<br>• Requires more computational resources.<br>• Issue with generalization. |
| Decision Tree | • Enable graphical representation.<br>• Highly interpretable<br>• No dependency on background knowledge.<br>• Highly interpretable.<br>• Ability to process data with multi-class classification.<br>• Not sensitive to abnormal and missing values.<br>• Minimal data pre-processing required. | • Have risk of overfitting (without pruning)<br>• Weaker performance as compared to ensemble learning and regression analysis.<br>• High variance. |
| SVM | • Suitable for small sample.<br>• Not sensitive to outliers.<br>• Robust to overfitting.<br>• Clear decision boundaries.<br>• Effective with small datasets.<br>• High accuracy for classification. | • Difficult to train using big dataset.<br>• Have difficulty with multi-class problem.<br>• Model performance is dependent on parameter selection.<br>• Computational complexity.<br>• Scalability issues.<br>• Sensitive to noise.<br>• Less interpretable models. |
| APCM | • Interpretable with result visualization.<br>• A training free method useful for small sample.<br>• Handles large and complex datasets.<br>• Identifies subtle patterns and anomalies in patients. | • The method is dependent on the accuracy of pose estimation of input data (video).<br>• Computational complexity. |
| Ensemble learning | • The performance of the model is improved to a certain extent compared with the weak classifier.<br>• Insensitive to outliers.<br>• High performance on large samples.<br>• It can deal with nonlinear problems.<br>• Little possibility of over fitting<br>• Robust to variations in data. | • The model is difficult to explain, and there is a black box problem.<br>• Normalization is required.<br>• Some models are sensitive to missing values.<br>• Complex to understand and implement.<br>• Requires more computational resources. |

*(Continued)*

| Algorithms | Advantages | Disadvantages |
|---|---|---|
| Associate rule | • The algorithm principle is simple and easy to implement.<br>• It is not restricted by dependent variables, and the association between data can be found in big data.<br>• Can be integrated with other machine learning methods. | • There are many output rules and a lot of useless information |
| Clustering | • Data driven insights.<br>• Can be easily integrated with other machine learning algorithms.<br>• Be able to handle big data problems.<br>• Strong interpretability. | • The model is sensitive to outliers.<br>• The model is sensitive to unbalanced data.<br>• Local optimal solutions are often obtained.<br>• Require high quality large datasets. |
| Dimensionality Reduction | • The model is fast, simple and effective.<br>• Noise reduction.<br>• Reduces risk of overfitting. | • Poor interpretability of the model.<br>• Dependency on data quality.<br>• Generalization issue. |
| CIMA | • Can reflect complexity and variability of infant spontaneous movements.<br>• Comparable accuracy with GMA and neonatal cerebral imaging. | • Dependency on quality and consistency of video data.<br>• Technical complexity—equipment and software setup for data capturing.<br>• Limited generalization.<br>• Delivers higher numbers of false positives. |
| GCN | • Higher accuracy than ML method (CIMA).<br>• Effective at handling complex data structures.<br>• Enables integration of multi-modal data.<br>• Scalable.<br>• Interpretable. | • Model complexity.<br>• Requires large amount of high-quality data.<br>• Computationally intensive training and deployment. |
| RUSBoost | • Handling of imbalanced data.<br>• Improved accuracy.<br>• Enhanced sensitivity and specificity.<br>• Scalability. | • Loss of information due to under sampling.<br>• Complex computation.<br>• Overfitting risk.<br>• Requires careful tuning of parameters.<br>• Dependency on quality of data. |
| CNN | • Higher accuracy for image classification.<br>• Handles complex and high-dimension data.<br>• Scalable.<br>• Higher sensitivity and specificity. | • High computational cost.<br>• Requires good quality of large datasets.<br>• Risk of overfitting.<br>• Complexity in interpretation. |
| LSTM | • Effective in handling sequential data.<br>• Higher accuracy with large data.<br>• Adaptability to different datasets.<br>• Scalable.<br>• Learn directly from raw statistical data. | • Computationally expensive.<br>• Requires more training time.<br>• Requires large amount of labelled data.<br>• Prone to overfitting.<br>• Lack interpretability. |
| FCNet | • Simple and efficient.<br>• Less prone to overfitting.<br>• More interpretable as compared to other deep learning models. | • Limited handling of sequential data.<br>• Lack of spatial awareness. |

| Table 6 (continued) | | |
|---|---|---|
| **Algorithms** | **Advantages** | **Disadvantages** |
| kNN | • Simple to implement.<br>• Flexibility with data types.<br>• Effective with small datasets.<br>• Interpretable. | • Computational complexity with large datasets.<br>• Sensitivity to irrelevant features.<br>• Accuracy depends on choice of k.<br>• Sensitive to noise.<br>• Requirement to store entire training dataset lead to high memory usage. |
| RNN | • Effective at processing sequential data.<br>• Ability to learn and recognize complex patterns in data.<br>• Can be combined with other machine learning models and data types. | • Computationally intensive.<br>• Vanishing gradient problems.<br>• Requires large amount of high quality and labelled data.<br>• Less interpretability. |
| MLP | • Performs better with smaller datasets. | • Limited handling of sequential and time-series data.<br>• Overfitting risk.<br>• Scalability issue. |

**Note:**
CIMA, Computer-based infant movement assessment; ML, machine learning; SVM, support vector machine 3; kNN, k-nearest neighbor; GCN, graph convolutional network; APCM, affinity propagation clustering model; CNN, convolutional neural network; LSTM, long short term Memory; FCNet, fully connected neural network; MLP, multi layer perceptron; RNN, recurrent neural network; CP, cerebral palsy.

**Areas in CP care where AI can be implemented**

**Diagnosis and assessment**

Early Detection
Dyskinetic CP detection
Predict brain age
Gait Analysis
Home based dystonia assessment
Automate posture identification
Early prediction of motor impairment
Identifying factors related to intellectual disabilities
Identifying treatment non-responders

**Classification**

Predict gross motor function
Physical activity classification
CP classification

**Management**

Mobility Aid
Pre-surgical planning

**Figure 8** Areas of CP patient care where research has been carried out for implementation of AI.

The research uses pose estimation algorithms that use a video and convert it into dots and lines. The algorithm makes simple and real-time visualizations of movements by the patient for doctors to evaluate remotely. Testing is being done on how machine learning can be used to apply the clinical scores by using the above information (*Ellis, 2023*).

Figure 8 highlights some of the areas in cerebral palsy diagnosis and treatment where research is being conducted to use artificial intelligence for better results.

### AI for early prediction of cerebral palsy in infants

Predicting CP early on is crucial for initiating timely interventions and enhancing outcomes, and machine learning (ML) and deep learning have emerged as valuable tools in this endeavor. The General Movements Assessment (GMA) method, introduced by *Einspieler et al. (1997)*, is a widely utilized conventional technique for early CP detection in infants. Experts analyze the infants' spontaneous movements to predict the likelihood of cerebral palsy. There is a growing interest in leveraging technology to aid clinical decision-making, overcome logistical challenges, enhance predictive accuracy, and facilitate early interventions.

Machine learning algorithms possess the capability to analyze large datasets and identify intricate patterns and anomalies. This capacity enables the early detection of diseases, leading to prompt interventions and better patient outcomes. Researchers have investigated a variety of machine learning methods, such as support vector machine (SVM), linear discriminant analysis (LDA), decision tree (Tree), logistic regression (LR), k-nearest neighbor (kNN), ensemble of classification models (Ens), and the Computer-based Infant Movement Assessment (CIMA) model, among others, to predict cerebral palsy early on This review provides an overview of several studies employing these methods.

The research discussed by *Migliorelli et al. (2020)* investigates the effectiveness of five widely-used machine learning classification algorithms: K-Star, multilayer perceptron (MLP), naïve Bayes (NB), random tree (RT), and support vector machine (SVM). Notably, the MLP classifier demonstrates a remarkable accuracy rate of 84% in recognizing cases of cerebral palsy and 53% in forecasting Gross Motor Function Classification System (GMFCS) levels. While the research covers various machine learning algorithms, it underscores the importance of carefully selecting features. The study suggests that further investigation into feature selection methods and interpretability techniques is necessary to advance our comprehension of the factors influencing CP classification.

In the study by *McCay et al. (2022)*, the authors propose an innovative approach to predict cerebral palsy (CP) in infants using video-based assessments. Unlike traditional classification methods, their method reframes the problem as a clustering task. They extract joint information from infant pose estimation and segment the skeleton sequence into clips. By quantifying the number of cluster classes, the proposed method achieves state-of-the-art performance on two datasets. Importantly, it offers interpretability and continuous quantification of infant brain development, advancing automatic health assessment for infants. While the proposed method achieves impressive results, it may still struggle with intra-class variation. Infants exhibit diverse spontaneous movements, and

accounting for this variability remains a challenge. Variations in illumination, camera motion, changes in subject scale, and inconsistencies in resolution across recorded video footage can introduce noise and impact the method's resilience. The article acknowledges that the method is limited by small samples.

In a study by *Hesse et al. (2019)*, a framework was proposed for diagnosing cerebral palsy using pose data extracted from standard 2D RGB video. The framework integrates feature extraction, feature fusion, and classification methods while prioritizing human interpretability throughout the classification process. This research focuses on the early diagnosis of cerebral palsy, an area that has received considerable attention from various disciplines recently. Although diagnostic tools like the GMA have demonstrated promising outcomes, automating these procedures can improve accessibility and comprehension of infant movement development.

The article by *Raghuram et al. (2022)* introduces an innovative approach for predicting CP risk in very preterm infants. The study leverages 2D video-based analysis, aiming to automate the assessment process and enhance early intervention strategies. In this ambispective cohort study, infants born at less than 31 weeks of gestational age (GA) were evaluated using the GMA. Instead of relying on extensive manual training for GMA, the authors propose an automated movement analysis method based on 2D video data. Through the examination of features such as mean vertical velocity, motion quantity, and variability, researchers have constructed a statistical model for forecasting the risk of cerebral palsy. The findings reveal encouraging specificity and negative predictive value, suggesting the potential utility of this technology as a screening tool for extremely premature infants. Nonetheless, additional validation in preterm and high-risk term populations is imperative to evaluate its clinical utility comprehensively. In summary, this study makes strides in enhancing the early prediction of cerebral palsy, potentially enhancing outcomes for affected infants.

*Morais et al. (2023)* introduced a technique named FidgetyFind to evaluate the quality of general movements in infants. These fidgety movements, observed between 9 to 20 weeks post-term, serve as a robust indicator of cerebral palsy. Unlike traditional methods that rely on complex models, FidgetyFind is training-free, interpretable, and accurate. It detects fidgety movements by measuring the movement direction variability of specific joints in short video segments. The method translates qualitative expert assessments into a fine-grained scoring system, closely resembling the domain expert process. Evaluated on a large clinical dataset, FidgetyFind outperforms many existing methods in terms of interpretability and accuracy. However, this method has certain limitations. It focuses specifically on detecting fidgety movements in infants. While these movements are highly indicative of cerebral palsy, they represent only a subset of an infant's overall motor behavior. The method may miss other relevant movement patterns that could contribute to a more comprehensive assessment of an infant's health. Also, it relies on video data to extract movement information. The accuracy of movement detection heavily depends on the quality of the recorded videos.

The research described by *Ihlen et al. (2019)* introduces the CIMA model, a novel machine-learning technique for the early prediction of cerebral palsy (CP) using video

recordings of infants. This model analyzes time-frequency decompositions of infant body part trajectories to quantify the percentage of movements associated with CP risk. Developed and evaluated on video recordings from 377 high-risk infants aged 9–15 weeks (corrected age), the model predicts CP status and motor function (ambulatory *vs.* non-ambulatory) at an average age of 3.7 years. Notably, the CIMA model exhibits accuracy comparable to the GMA and neonatal cerebral imaging. Additionally, it effectively distinguishes between children with ambulatory and non-ambulatory CP. However, the implementation of the CIMA model in clinical settings presents practical challenges.

*Rahmati et al. (2015)* introduces an innovative method for predicting CP in infants using motion data. Instead of relying on traditional features in the time domain, the study proposes a collection of features obtained from frequency analysis of infants' movements. Given that cerebral palsy impacts motion variability, frequency analysis aligns closely with the condition's characteristics. To tackle the challenge of limited subjects and numerous features, the authors propose a feature selection technique that identifies relevant features with substantial predictive capability. This method reduces the risk of false discoveries, thereby enhancing the validity and applicability of the prediction model. The attained sensitivity of 86%, specificity of 92%, and accuracy of 91% demonstrate favorable performance compared to advanced clinical techniques for predicting cerebral palsy. Although this method shows promise, there are certain limitations, such as obtaining high-quality motion data and understanding the clinical relevance of specific frequency components, *etc.*, for its clinical implementation.

Deep learning model implementations in CP early predictions have shown promising results. *Groos et al. (2022)* leverages deep learning techniques to predict CP risk based on video recordings of infants' spontaneous movements. Deep learning models have shown remarkable success in various domains, and their application to CP prediction is promising. The method achieves a specificity of 94%, indicating its ability to identify infants without CP correctly. This specificity is crucial for minimizing false positives and ensuring that healthy infants are not unnecessarily flagged for further evaluation. The model's performance is comparable to that of existing clinical methods such as the General Movement Assessment (GMA) and neonatal imaging. While the model performs well in research settings, translating it into clinical practice poses challenges. Clinicians must understand the model's predictions, interpret its features, and integrate it seamlessly into diagnostic workflows.

*Devarajan & Khader (2023)* introduces an innovative approach to enhance the detection of CP using generative adversarial networks (GANs). The authors address the scarcity of annotated infant movement data by proposing a pose sequence-aware GAN (PS-GAN)-based data augmentation method. First, the PS-GAN captures long-range dependencies in continuous frames through self-attention and prunes the dense graph for efficient training. Next, spatial joints and temporal characteristics are encoded into the PS-GAN using graph convolutional networks (GCNs), resulting in high-quality skeleton images. The article also defines the PS-GAN structure selection as a Markov decision process (MDP) and solves it using reinforcement learning (RL). While the PS-GAN achieves impressive accuracy in CP detection, the article lacks detailed insights into the

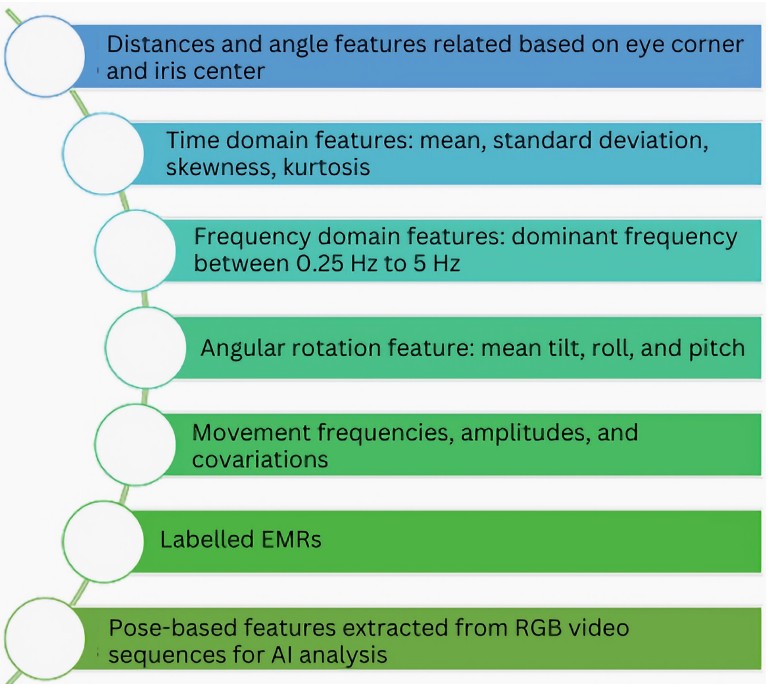

**Figure 9** Some common features extracted from data for AI-based CP patient care.

model's decision-making process. There remains a gap in understanding which features or patterns contribute to accurate predictions. Figure 9 highlights some of the features extracted in studies related to cerebral palsy patient care using artificial intelligence.

### AI for classification and treatment of cerebral palsy

Various machine learning and deep learning algorithms have been studied and implemented by researchers to diagnose various conditions associated with cerebral palsy (*Pham et al., 2021*; *Crowgey et al., 2018*; *MacWilliams et al., 2022*; *Zhang & Ma, 2019*; *Chakraborty & Nandy, 2020*). The implementation of AI for CP classification based on GMFCS is studied by many researchers to facilitate the best treatment plans and outcomes (*Schafmeyer et al., 2024*; *Duran et al., 2022*; *von Elling-Tammen et al., 2023*; *Ahmadi et al., 2020*; *Bertoncelli et al., 2019*; *Hou et al., 2023*). Apart from this, research on the use of AI for rehabilitation is an area with great potential.

*Illavarason, Arokia Renjit & Mohan Kumar (2019)* propose a computational approach to automatically assess the progress of children with cerebral palsy (CP) and evaluate their performance. The study utilizes eye movement data from 40 CP children (aged 3–11 years) with relatively mild motor impairment. Through the analysis of abnormal eye conditions using machine learning classification algorithms, the method achieves a peak classification accuracy of 94.17% with a neural network classifier. Specificity and sensitivity rates are also documented as 0.9800 and 0.9165, respectively. This research contributes to the non-invasive and precise detection of abnormalities in children with CP, thereby assisting in their rehabilitation.

In the study *von Elling-Tammen et al. (2023)*, the authors aimed to forecast the Gross Motor Function Measure-66 (GMFM-66) score using medical devices employed by patients with cerebral palsy (CP). They devised the Medical Device Score Calculator (MDSC) based on data from 1,581 children and adolescents with CP. Among various machine learning algorithms, the random forest algorithm exhibited the highest accuracy, with a concordance correlation coefficient (Lin) of 0.75. The MDSC is suitable for scientific applications, such as comparing or evaluating the effectiveness of different therapies, but not for individual patient assessments.

Studies have been conducted to examine various features using machine learning for the classification of CP. In one study (*Al-Sowi et al., 2023*), researchers assembled a comprehensive dataset for Jordanian children with CP. They assessed this dataset using five machine learning algorithms: Random Tree (RT), Naïve Bayes (NB), K-Star, Multilayer Perceptron (MLP), and Support Vector Machine (SVM). The MLP classifier achieved an accuracy of 84% in CP-type classification and 53% in the Gross Motor Function Classification System.

Researchers have also endeavored to investigate the gait patterns of individuals to design effective therapies for them. In the study *Kuntze et al. (2018)*, k-means clustering was employed to analyze the barefoot walking kinematics of 37 male and female children and youth with spastic diplegic cerebral palsy. They identified up to four kinematic clusters based on multi-joint angles without prior data reduction. These clusters provided insights into distinct gait patterns, potentially improving clinical management for individuals with cerebral palsy. The Silhouette value demonstrated a cluster boundary effect, indicating that data with values approaching zero were more likely to change cluster allocation.

Diagnostic techniques such as functional MRI (fMRI) are employed in the diagnosis of CP (*Reid et al., 2016*; *Palraj & Siddan, 2021*). In the study (*Reid et al., 2016*), researchers endeavor to classify cerebral palsy using fMRI images. They propose a deep convolutional network based on a modified AlexNet architecture to differentiate between different types of cerebral palsy based on fMRI brain images. This methodology aids physicians in devising effective rehabilitation strategies for children affected by the condition. Lower limb rehabilitation for cerebral palsy mostly focuses on conventional physiotherapy sessions that cover exercises to increase strength, balance, and coordination (*Tunde Gbonjubola, Garba Muhammad & Tobi Elisha, 2021*; *Das & Ganesh, 2019*; *Patel et al., 2020*). These exercises are labor intensive, and the outcome is qualitative. Recently, various researchers and engineers have been working on lower limb exoskeleton models to support patients with cerebral palsy for walking (*Hegde et al., 2018*; *Sarajchi, Al-Hares & Sirlantzis, 2021*; *Diot et al., 2023*; *Bunge et al., 2021*; *Orekhov et al., 2020*; *Lerner, Damiano & Bulea, 2017*; *Lerner et al., 2017*). In the study (*Luo et al., 2023*), researchers have developed a novel controller for lower limb rehabilitation exoskeletons (LLREs). Utilizing deep neural networks and reinforcement learning, the controller seeks to offer dependable walking assistance by managing uncertain forces during human-exoskeleton interaction. Trained through a decoupled offline simulation of human-exoskeleton dynamics, the controller employs three distinct networks, eliminating the need for control parameter adjustments and enabling support for individuals with diverse neuromuscular conditions.

*Kolaghassi, Marcelli & Sirlantzis (2023)* explore the utilization of deep learning algorithms to forecast stable gait trajectories in children diagnosed with cerebral palsy. Advanced models, including transformers, long short-term memory networks, convolutional neural networks, and fully connected neural networks, were utilized to forecast patterns in gait trajectory. The research zeroes in on exoskeleton reference trajectories and their potential application in aiding children with neurological conditions.

*Zhang & Ma (2019)* assessed various machine learning algorithms to classify gait patterns in children afflicted with cerebral palsy, specifically those with spastic diplegia. They extracted gait parameters from data collected from 200 children with spastic diplegia CP, utilizing these parameters to represent key kinematic aspects of each individual's gait. The study compared seven supervised machine learning algorithms: discriminant analysis, naive Bayes, decision tree, k-nearest neighbors (KNN), support vector machine (SVM), random forest, and artificial neural network (ANN). The ANN demonstrated the highest prediction accuracy (93.5%) and a low resubstitution error, suggesting its promise for classifying gait in children with spastic diplegia CP. The decision tree algorithm also displayed promise due to its transparency for clinical utilization.

Dystonia, characterized by involuntary muscle contractions and movements, can affect a specific body part or the entire body, falling under the spectrum of dyskinetic cerebral palsy (*Monbaliu et al., 2017*; *Sanger, 2015*). *Haberfehlner et al. (2023)* focuses on automating the video-based evaluation of dystonia in dyskinetic cerebral palsy using machine learning methods. The authors introduce a novel approach for assessing dystonia in dyskinetic CP, extracting 2D stick figure data from videos *via* markerless motion tracking (*i.e.*, x, y coordinates of body parts). Supervised machine learning techniques are subsequently utilized to forecast dystonia scores, utilizing computed movement and positional features obtained from these coordinates. This approach aims to automate the assessment process, which currently relies heavily on clinician expertise and is time-consuming. Figure 10 shows the overview of different AI classifiers used for research in Cerebral Palsy patient care.

Table 7 summarizes the datasets used, AI models, and their measured outcomes and results in a few of the studies in the domain of AI applications in CP patient care.

### Evaluation metrics used

In the realm of AI applications in CP patient care, research spans various domains, such as early prediction, classification, and treatment, among others. Evaluation metrics employed to gauge research outcomes vary based on the specific application area. Parameters such as the Area Under the Receiver Operating Characteristic Curve (AUC-ROC), decision threshold, classification uncertainty, MCC (Matthew's correlation coefficient) (Eq. (6)), Gross Motor Function Classification System (GMFCS), aggregated CP risk, true positive (TP), true negative (TN), false positive (FP), false negative (FN), sensitivity (SE), specificity (SP), and accuracy (AC), precision (PR), recall (RE) (Eq. (5)), F Score (F) (Eq. (6)), mean squared error (Eq. (8)) (MSE), and mean average error (MAE) (Eq. (9)) are among those utilized to assess the efficacy of AI interventions in CP patient care.

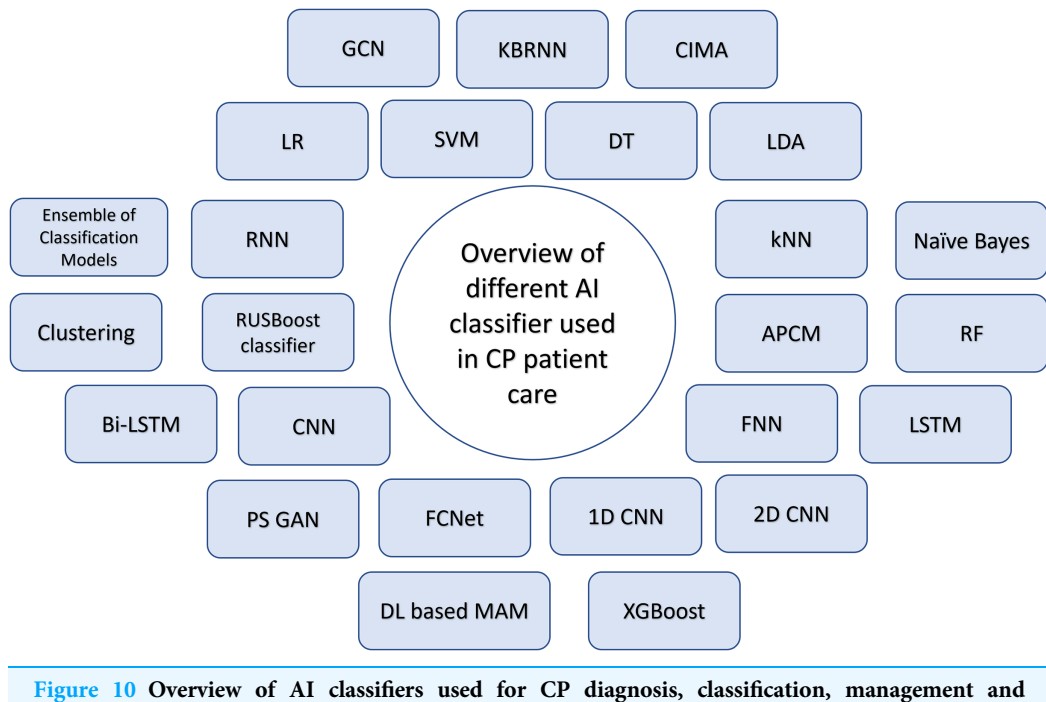

**Figure 10 Overview of AI classifiers used for CP diagnosis, classification, management and treatment.**

As described in Eq. (1), sensitivity assesses the proportion of correctly identified positive classifications among the positive dataset population. In contrast, specificity, outlined in Eq. (2), gauges the percentage of correctly identified negative classifications within the negative dataset population. Accuracy, as represented by Eq. (3), indicates the proportion of all instances correctly classified. Precision, as illustrated in Eq. (4), represents the percentage of accurately identified positive cases out of all positive predictions. At the same time, recall signifies the correctly identified positive cases among all actual positive cases.

Here are the formulas to calculate some of the most commonly used metrics:

$$SE = \frac{TP}{TP + FN} \tag{1}$$

$$SP = \frac{TN}{TN + FP} \tag{2}$$

$$AC = \frac{TP + TN}{TN + FN + TP + FP} \tag{3}$$

$$PR = \frac{TP}{TP + FP} \tag{4}$$

$$RE = \frac{TP}{TP + FN} \tag{5}$$

$$MCC = \frac{TP \times TN - FP \times FN}{\sqrt{(TP + FP)(TP + FN)(TN + FP)(TN + FN)}} \tag{6}$$

where TP—Correctly classified as impaired/affected/diseased.

TN—Correctly classified as not impaired/unaffected/not diseased.

**Table 7 Summary of datasets, AI models their measured outcomes and results in few of the studies in the domain of AI applications in CP patient care.**

| Ref | Year | Data | Classifier | Evaluation metrics | Findings | Results |
|---|---|---|---|---|---|---|
| Li et al. (2022) | 2022 | Patient data from EMR records | KBRNN | Diagnostic accuracy | KBRNN outperforms traditional neural network models in syndrome diagnosis tasks. | KBRNN<br>Diagnostic accuracy with knowledge injection - 79.31%<br>Fully trained KBRNN accuracy in syndrome diagnosis - 83.12% |
| Ihlen et al. (2019) | 2020 | Infant video recordings | CIMA | Sensitivity, specificity, PPV, NPV, AUC | CP prediction performance depends on the decision threshold for CP risk-related movements.<br><br>CIMA model predicts CP in high-risk infants with accuracy and it differentiates between ambulatory and non-ambulatory CP. | CIMA<br>• Sensitivity - 92.7%<br>• Specificity - 81.6%<br>• Positive predictive value - 38.0%<br>• Negative predictive value - 98.9%<br>• AUC - 0.87<br><br>GMA<br>• Sensitivity - 76.2%<br>• Specificity - 82.4%<br>• Positive predictive value - 33.3%<br>• Negative predictive value - 96.8%<br>• AUC - 0.82<br><br>Imaging<br>• Sensitivity - 81.0%<br>• Specificity - 85.3%<br>• Positive predictive value - 39.1%<br>• Negative predictive value - 97.5%<br>• AUC - 0.85<br><br>$C_{SD}$<br>• Sensitivity - 56.1%<br>• Specificity - 58.6%<br>• Positive predictive value - 14.2%<br>• Negative predictive value - 91.6%<br>• AUC - 0.56 |

(Continued)

| Ref | Year | Data | Classifier | Evaluation metrics | Findings | Results |
|---|---|---|---|---|---|---|
| *Groos et al. (2022)* | 2022 | Infant video recordings | DL-GCN ML- CIMA | Sensitivity, specificity, PPV, NPV, accuracy | Deep learning methods predicted CP with high sensitivity and specificity. Achieved higher accuracy than conventional machine learning methods.<br><br>Differentiated between ambulatory *vs.* nonambulatory CP and spastic unilateral *vs.* bilateral. | Deep learning based method-<br>• Sensitivity - 71.4%<br>• Specificity - 94.1%<br>• Positive predictive value - 68.2%<br>• Negative predictive value - 94.9%<br>• Accuracy - 90.6%<br><br>GMA-<br>• Sensitivity - 70.0%<br>• Specificity - 88.7%<br>• Positive predictive value - 51.9%<br>• Negative predictive value - 94.4%<br>• Accuracy - 85.9%<br><br>ML methods-<br>• Sensitivity - 71.4%<br>• Positive predictive value - 31.9%<br>• Negative predictive value - 93.5%<br>• Accuracy - 72.7%<br>• Specificity - 72.9% |
| *McCay et al. (2022)* | 2022 | MINI-RGBD RVI-38 | LR, SVM, DT, LDA, Ensemble of classification models, kNN | Accuracy, F1 score, MCC | The individual characteristics exhibited comparable performance to the top features suggested in similar studies. Yet, the amalgamation of these proposed attributes offers cutting-edge performance and interpretability | **Performance of proposed features**<br>• Accuracy: 92.11%<br>• F1 score: 76.92%<br>• Matthews correlation coefficient (MCC): 72.51%<br><br>**The HOAD2D feature**<br>• F1 Score: 83.33%<br>• MCC: 80.21%<br><br>**The HORJO2D feature**<br>• Classification accuracy: 91.67%<br>• F1 score: 85.71%<br>• MCC: 81.65%<br><br>**The HORJAD2D feature**<br>• Classification accuracy: 83.33%<br>• F1 Score: 75%<br>• MCC: 62.50% |

| Ref | Year | Data | Classifier | Evaluation metrics | Findings | Results |
|-----|------|------|------------|--------------------|---------|---------|
| *Raghuram et al. (2022)* | 2022 | Preterm infants video data | Multivariable LR | Sensitivity, specificity, PPV, NPV | Automated analysis predicts cerebral palsy with 55% sensitivity. | Multivariable logistic regression model—Automated movement analysis<br>• Sensitivity - 55.17%<br>• Specificity - 79.64%<br>• Positive predictive value - 26.23%<br>• Negative predictive value - 93.12%<br><br>Multivariable logistic regression model—Clinical GMA<br>• Sensitivity - 85.71%<br>• Specificity - 96.43%<br>• Positive predictive value - 66.67%<br>• Negative predictive value - 98.78% |
| *McCay et al. (2020)* | 2021 | Neonates EEG data | RUSBoost classifier | Sensitivity, specificity, classification accuracy, AUC | EEG complexity features and graph-theoretic parameters are potential CP biomarkers. | Classification accuracy (AAC) - 84.6%<br>Sensitivity (SNS) - 83%<br>Specificity (SPC) - 85%<br>Area under Curve (AUC) - 0.87 |
| *Bakheet et al. (2021)* | 2023 | Preterm infants MRI data | Semi supervised GCN | Sensitivity, specificity, BA, accuracy, AUC | The GCN model achieved an accuracy of 66.4% and an AUC of 0.67 with only labeled data. With additional unlabeled data, the accuracy improved to 68.0% ($p = 0.016$) and the AUC increased to 0.69 ($p = 0.029$). | (GCN) With only labelled data-<br>• Sensitivity - 61.7%<br>• Specificity - 69%<br>• Balanced accuracy (BA) - 65.4%<br>• Accuracy - 66.4%<br>• AUC - 0.67<br><br>(GCN) With labelled and unlabelled data<br>• Sensitivity - 63.1%<br>• Specificity - 70.2%<br>• Balanced accuracy (BA) - 66.7%<br>• Accuracy - 68%<br>• AUC - 0.69<br><br>Logistic regression<br>• Sensitivity - 51.6%<br>• Specificity - 65.7%<br>• Balanced accuracy (BA) - 58.7%<br>• Accuracy - 60.1%<br>• AUC - 0.59 |

(Continued)

| Ref | Year | Data | Classifier | Evaluation metrics | Findings | Results |
|-----|------|------|------------|--------------------|----------|---------|
| | | | | | | Ridge classifier<br>• Sensitivity - 50.5%<br>• Specificity - 71.9%<br>• Balanced accuracy (BA) - 61.2%<br>• Accuracy - 63.3%<br>• AUC - 0.62<br><br>SVM (linear kernel)<br>• Sensitivity - 59.8%<br>• Specificity - 69.1%<br>• Balanced accuracy (BA) - 64.5%<br>• Accuracy - 65.7%<br>• AUC - 0.64<br><br>SVM (rbf kernel)<br>• Sensitivity - 61.5%<br>• Specificity - 68.2%<br>• Balanced accuracy (BA) - 64.9%<br>• Accuracy - 66.2%<br>• AUC - 0.65<br><br>Deep neural network<br>• Sensitivity - 60.2%<br>• Specificity - 68.9%<br>• Balanced accuracy (BA) - 64.6<br>• Accuracy - 65.9%<br>• AUC - 0.64 |
| *Wu et al. (2023)* | 2023 | MINI RGBD RVI-38 | APCM | Sensitivity, specificity, accuracy | The method suggested accurately measures atypical brain growth in babies and can be applied to various data collections without requiring training. | Using MINI RGBD dataset<br>• Accuracy - 91.67%<br>• Sensitivity - 100%<br>• Specificity - 87.5%<br><br>Using RVI 38 dataset<br>• Accuracy - 94.74%<br>• Sensitivity - 83.33%<br>• Specificity - 96.88% |

| Ref | Year | Data | Classifier | Evaluation metrics | Findings | Results |
|---|---|---|---|---|---|---|
| *Sakkos et al. (2021)* | 2021 | MINI RGBD RVI 25 | CNN LSTM | Sensitivity, specificity, accuracy, precision, F1 score | Proposed framework identifies fidgety movements in cerebral palsy prediction. The Limb-based 8 joint variant shows the best performance in CP prediction. | Using MINI RGBD dataset—Proposed method<br>• Accuracy - 1.000<br>• Sensitivity - 1.000<br>• Specificity - 1.000<br>• Precision - 1.000<br>• F1 score - 1.000<br><br>Using RVI 25 dataset—Proposed method<br>• Accuracy - 0.920<br>• Sensitivity - 0.833<br>• Specificity - 0.947<br>• Precision - 0.833<br>• F1 score - 0.833 |
| *Mathis et al. (2018)* | 2023 | MINI RGBD Babypose MIA | PS GAN CNN | Accuracy | PS-GAN-CNN achieves 92.2%, 92.5%, and 92% accuracy in databases. | Accuracy<br>• MINI RGBD - 92.2%<br>• On BabyPose - 92.5%<br>• On MIA - 92% |
| *van der Krogt & Haberfehlner (2021)* | 2020 | MINI RGBD | FCNet | Classification accuracy | The FCNet proposed exhibited strong and consistent performance across various feature sets. | Average classification accuracy using HOJO2D feature set:<br>FCNet - 83.33%<br>1D CNN - 1 - 79.17%<br>1D CNN - 2 - 80.55%<br>2D CNN - 1 - 79.17%<br>2D CNN - 2 - 81.94% |
| | | | 1D - CNN | | Convolutional neural networks showcased outstanding capabilities in managing features with higher dimensionality. | Average classification accuracy using HOJD2D feature set:<br>FCNet - 86.11%<br>1D CNN - 1 - 81.94%<br>1D CNN - 2 - 81.94%<br>2D CNN - 1 - 81.94%<br>2D CNN - 2 - 80.55% |

*(Continued)*

| Ref | Year | Data | Classifier | Evaluation metrics | Findings | Results |
|-----|------|------|-----------|--------------------|----------|---------|
| | | | 2D - CNN | | Deep learning frameworks displayed reduced sensitivity to alterations in hyperparameters. | Average classification accuracy using fusion of HOJO2D and HOJD2D feature sets: FCNet - 83.33% 1D CNN - 1 - 84.72% 1D CNN - 2 - 90.28% 2D CNN - 1 - 79.17% 2D CNN - 2 - 80.55% |
| Rahmati et al. (2016) | 2016 | Infants' sensors and video data | PLSR (Statistical model), classification | Sensitivity, specificity, accuracy, AUC | Feature reduction method is useful to improve performance with classification model for sensor dataset. Promising performance was seen using the proposed features with video dataset. | Model predicts cerebral palsy in infants using motion data features. Achieved sensitivity - 86% Specificity - 92% Accuracy - 91% |
| Hartog et al. (2022) | 2022 | IMU data | DT, DA, NB, SVM, kNN, Ensemble learning | F1 score | It is feasible to collect data in natural environment for dystonia in CP using sensors. | • Best algorithm for dystonia of lower extremity: ENS <br>• Best algorithm for dystonia of upper extremity: KNN <br>• RMSE for dystonia of lower extremities: 1.07 <br>• RMSE for dystonia of upper extremities: 0.98 <br><br>For all best models combined: Dystonia of lower extremity: <br>• Mean F1 score validation: 0.97 <br>• Mean F1 score test: 0.67 <br>• Mean precision test: 0.82 <br>• Mean recall test: 0.66 <br><br>Dystonis of upper extremity <br>• Mean F1 score validation: 0.93 <br>• Mean F1 score test: 0.68 <br>• Mean precision test: 0.73 <br>• Mean recall test: 0.66 |

| Ref | Year | Data | Classifier | Evaluation metrics | Findings | Results |
|---|---|---|---|---|---|---|
| *Morbidoni et al. (2021)* | 2021 | sEMG data | SVM, RF, kNN, MLP, RNN | MAE, F1 score, TD, accuracy, precision, recall | Neural networks can predict 2 important gait events using sEMG. | Intra-subject<br>• Mean classification Accuracy - 0.97 ± 0.01<br><br>Inter-subject<br>• Mean classification Accuracy - 0.91 ± 0.03<br><br>Average MAE for heel strike - 14.8 ± 3.2 ms<br>Average MAE for toe-off - 17.6 ± 4.2 ms<br>F1 score for heel strike - 0.95 ± 0.03<br>F1 score for toe-off - 0.92 ± 0.07 |
| *Li et al. (2023)* | 2023 | Infants MRI data | Semi-supervised GCN | SD, accuracy, balanced accuracy, sensitivity, specificity, AUC | Semi-supervised GCN performs better than many supervised learning models to predict CP in infants. | (GCN) With only labelled data-<br>• Sensitivity - 61.7%<br>• Specificity - 69%<br>• Balanced accuracy (BA) - 65.4%<br>• Accuracy - 66.4%<br>• AUC - 0.67<br><br>(GCN) With labelled and unlabelled data<br>• Sensitivity - 63.1%<br>• Specificity - 70.2%<br>• Balanced accuracy (BA) - 66.7%<br>• Accuracy - 68%<br>• AUC - 0.69<br><br>Logistic regression<br>• Sensitivity - 51.6%<br>• Specificity - 65.7%<br>• Balanced accuracy (BA) - 58.7%<br>• Accuracy - 60.1%<br>• AUC - 0.59 |

(Continued)

| Ref | Year | Data | Classifier | Evaluation metrics | Findings | Results |
|---|---|---|---|---|---|---|
| | | | | | | Ridge classifier<br>• Sensitivity - 50.5%<br>• Specificity - 71.9%<br>• Balanced accuracy (BA) - 61.2%<br>• Accuracy - 63.3%<br>• AUC - 0.62<br><br>SVM (linear kernel)<br>• Sensitivity - 59.8%<br>• Specificity - 69.1%<br>• Balanced accuracy (BA) - 64.5%<br>• Accuracy - 65.7%<br>• AUC - 0.64<br><br>SVM (rbf kernel)<br>• Sensitivity - 61.5%<br>• Specificity - 68.2%<br>• Balanced accuracy (BA) - 64.9%<br>• Accuracy - 66.2%<br>• AUC - 0.65<br><br>Deep neural network<br>• Sensitivity - 60.2%<br>• Specificity - 68.9%<br>• Balanced accuracy (BA) - 64.6<br>• Accuracy - 65.9%<br>• AUC - 0.64 |
| *Ahmadi et al. (2018)* | 2018 | Sensor data (accelerometer) | BDT, RF, SVM | F-score, classification accuracy, | RF and SVM performs better than BDT to classify gait using sensor data. | Hip and wrist classifiers showed comparable prediction accuracy.<br><br>Combined hip and wrist classifiers achieved the best overall performance.<br><br>Recognition accuracy was excellent for sedentary activities.<br><br>ML methods provided acceptable classification accuracy for detecting various activities in CP. |

| Ref | Year | Data | Classifier | Evaluation metrics | Findings | Results |
|---|---|---|---|---|---|---|
| Ferrari et al. (2019) | 2019 | Optoelectronic markers and EMG data | MLP, LSTM, RNN | Accuracy | RNN and MLP are better than SVM where RNN performs better than MLP. | Accuracy achieved by SVM using stride length and cadence: 96.8%<br>Accuracy achieved by RNN: 0.869<br>Accuracy achieved by MLP: 0.804 |
| Cunningham et al. (2019) | 2019 | Video data | CNN | MAE, accuracy, precision, recall, False positive rates, false negative rates, F1-score | Deep learning can be used for automated posture identification in children with CP.<br>Demonstration of technical feasibility to automate the identification of sitting segmental posture. | The point-features were estimated with error $4.4 \pm 3.8$ pixels at approximately 100 images per second.<br>Truncal segment angles estimated with error $6.4 \pm 2.8°$.<br>Classification of deviation from reference posture with F1 > 80%. |
| Monica & Parvathi (2023) | 2023 | WISDM (Wireless sensor data mining) lab dataset | 2D CNN, LSTM, Bi-LSTM | Accuracy, loss, precision, | AI models can recognize the daily routine-based human activities with extreme accuracy. | **Models:**<br>2D CNN<br>• Accuracy: 86.92%<br>• Loss: 0.3532<br>LSTM<br>• Accuracy: 86.44%<br>• Loss: 0.9612<br>Bi-LSTM<br>• Accuracy: 87.47%<br>• Loss: 1.1653<br>**Evaualtion of gait parameters**<br>Stride length:<br>• Accuracy: 0.056<br>• Precision: 0.038 |

(Continued)

| Ref | Year | Data | Classifier | Evaluation metrics | Findings | Results |
|-----|------|------|------------|--------------------|----------|---------|
| | | | | | | Speed:<br>• Accuracy: 0.042<br>• Precision: 0.049<br><br>Strike angle:<br>• Accuracy: 0.56<br>• Precision: 2.16<br><br>Turning angle:<br>• Accuracy: 0.35<br>• Precision: 1.79<br><br>Lift-off angle:<br>• Accuracy: 4.28<br>• Precision: 5.68 |

**Note:**
EMR, Electronic Medical Record; KBRNN, Knowledge-based recurrent neural network; CIMA, Computer-based infant movement assessment; PPV, Positive Predictive Value; NPV, Negative Predictive Value; AUC, Area Under Curve; DL, Deep Learning; ML, Machine Learning; MINI-RGBD, Moving Infants in Red Green Blue Depth; RVI 38, Royal Victoria Infirmary 38; SVM, Support Vector Machine3; DT, Decision Tree; LDA, Linear Discriminant Analysis; LR, Logistic Regression; and kNN, k-Nearest Neighbor; MCC, Matthews Correlation Coefficient; EEG, Electroenchephalogram; GCN, Graph Convolutional Network; MRI, Magnetic Resonance Imaging; BA, Balanced Accuracy; APCM, Affinity propagation clustering model; RVI 25, Royal Victoria Infirmary 25; CNN, Convolutional Neural Network; LSTM, Long Short Term Memory; Bi-LSTM, Bidirectional long short term memory; PS-GAN-CNN, Pose Sequence Aware Generative Adversarial Network; FCNet, Fully Connected Neural Network; NB, Naïve Bayes; DA, Discriminant analysis; sEMG, Surface EMG; MLP, Multi layer perceptron; RF, Random Forest; MAE, Mean average error; TD, Time delay; RNN, Recurrent neural network; BDT, Binary Decision Tree.

FP—Unimpaired/unaffected/not-diseased incorrectly classified as impaired/affected/diseased.

FN—Impaired/affected/diseased incorrectly classified as unimpaired/unaffected/not-diseased.

$$F = \frac{2 \times PR \times RE}{PR + RE} \tag{7}$$

where,

PR—Precision
RE—Recall

$$MSE = \frac{1}{N} \sum_{i=1}^{N} (y_i - \widehat{y_i})^2 \tag{8}$$

where,

N = Total number of data points
$y_i - \hat{y}_i$ = Square of difference between actual and predicted value.

$$MAE = \frac{\sum_{i=1}^{n} |y_i - x_i|}{n} \tag{9}$$

where,

$y_i$ = prediction
$x_i$ = true value
n = total number of data points.

## RQ 4—CHALLENGES AND LIMITATIONS OF USING AI FOR CP PATIENT CARE

Although machine learning and deep learning techniques show significant promise, Researchers should address concerns related to scalability, cost-effectiveness, clinical acceptance, and resources needed for widespread adoption.

**Dataset challenges and patient privacy:** Some of the studies for early prediction of cerebral palsy in infants are related to the use of computer vision techniques, such as motion image generation through frame differencing, which can be sensitive to the movement of the camera, background pre-processing necessity, self-occlusion, *etc*. This may make it challenging to pre-process the data. Dependency on the hardware setup and specification in these cases is another limitation for implementing these models in primary care centers due to budgetary constraints.

One of the major challenges is the availability of the dataset used for the research. Most of the research in this field uses small individual datasets to produce the results. Hence, these studies reported performance evaluation parameters are difficult to compare, examine, and generalize. The models developed in these studies are not robust as the data used to train these models is limited. Also, CP is a neurological condition affecting neuromotor abilities; hence, the non-invasive data that we acquire for ML techniques also needs to be mapped and compared with invasive diagnosis methods carried out for the

patients. Study and comparison of electroencephalogram (EEG) and electromyography (EMG) signals need to be included in the studies in order to develop a more practical approach for the treatment of CP patients. In the studies related to CP diagnosis and prognosis, the limited number of patients, non-inclusivity of the EEG and EMG signals, and differences in the features used make it more complex to come to a conclusion about the effectivity, efficiency, and reliability of the techniques proposed. As far as the role of AI-ML is concerned in the treatment of CP, there is a high probability that these various feature-based studies can at least help to improve neuromotor coordination. Still, to increase the reliability and outcome, there is a need to have more public data sets and use various feature-based methods along with EEG and EMG signals for decision-making and training of AI-ML methods.

The process of aggregating or disclosing data faces obstacles not only due to regulatory, ethical, and legal issues concerning privacy and data security but also due to technical challenges. Safeguarding the privacy of healthcare data, managing access securely, and anonymizing information pose complex challenges, which can sometimes be insurmountable.

**Lack of interpretability:** Many studies on CP early prediction and image analysis have reported that Deep learning models lack interpretability, limiting the clinical adaptations of the models.

The interpretation and clinical applicability of the results pose a significant challenge when applying machine learning to healthcare. The complex nature of machine learning, particularly deep learning methods, makes it difficult to discern the original features' impact on predictions. While this may not be a major issue in other machine learning applications like web searches, the lack of transparency has become a major obstacle to integrating machine learning into healthcare. In healthcare, it's crucial to understand that the approach to finding a solution is just as important as the solution itself. There needs to be a deliberate shift towards identifying and quantifying the data features that drive predictions. Involving physicians and healthcare professionals in the development, implementation, and testing of machine learning methods may also help increase their acceptance and use.

**Feature extraction:** Understanding and extracting meaningful features from the pose data of infants is challenging due to their complex movement patterns. These movements are influenced by factors such as muscle tone, reflexes, and developmental stages, making them highly variable, especially in the early stages of development. Additionally, noise from camera sensors, lighting conditions, and occlusions complicates feature extraction, requiring robustness and consistency across different video sequences. When predicting cerebral palsy (CP), AI models must consider diverse clinical contexts, including factors like age, gestational history, and comorbidities, which impact predictive accuracy. Maintaining a balance between specificity (identifying true cases) and sensitivity (minimizing false negatives) is crucial. While models trained on controlled datasets perform well, deploying them in real-world scenarios, such as home environments,

remains challenging due to the need for adaptation to varying camera qualities, lighting, and background clutter for practical use.

## RQ 5—FUTURE DIRECTIONS

**Building new and bigger datasets:** The lack of larger and more diverse datasets needs to be addressed in order to conduct quality research in this field. It is important to build different types of datasets that cater to specific age groups and the intended uses of the data. Models should be developed to process this data while maintaining anonymity. Efforts should be made to annotate the datasets. The models developed using larger datasets can then be further tested and implemented in clinical settings.

**Incorporating model interpretability framework:** Medical diagnosis and treatment need reasoning and explanation for each step carried out during the process to assure reliability and patient safety. There's a drive towards enhancing the transparency of algorithms through efforts in explainable artificial intelligence (XAI). The objective of XAI is to develop models capable of elucidating their outputs, such as indicating the features they consider when making predictions. This endeavor aims to bolster transparency, thereby fostering greater trust and comprehension of AI predictions by humans. Certain AI techniques are inherently more conducive to explanation than others. Future research should prioritize the development and refinement of visualization tools that facilitate access to transparent and comprehensible decision-making processes.

**Feature selection:** Choosing an appropriate deep learning architecture is pivotal for feature extraction. Architectures such as convolutional neural networks (CNNs) excel at learning hierarchical features directly from raw image data, making them adept at extracting pertinent features from pose data. Studies have explored CNN-based architectures for inferring joint angles and scrutinizing movements. On the other hand, recurrent neural networks (RNNs) are adept at capturing temporal dependencies in sequential data, crucial for analyzing movement trajectories longitudinally. They are able to model the dynamic nature of infant movements and predict the risk of cerebral palsy. More research is needed to gather further evidence. Utilizing pretrained models through transfer learning, such as those from ImageNet, can help leverage knowledge from large datasets. Fine-tuning these models with CP-specific data can enhance feature extraction. Adapting models trained on synthetic or controlled datasets for real-world scenarios is essential. Techniques like domain adaptation and adversarial training can help bridge the gap between laboratory settings and clinical environments. Diagnosing cerebral palsy is a complex process that utilizes various modalities to ensure accurate diagnosis. This may involve imaging modalities like ultrasound and MRI, physical assessments, video-based assessments, EEG, and EMG data, among others. Features associated with these modalities vary. Integrating information from multiple sources, such as pose data, electromyography, and accelerometers, enhances feature extraction. Fusion techniques, whether early, late, or cross-modal, can enhance predictive accuracy. Attention mechanisms enable models to focus on relevant features within multimodal data, thus aiding in robust feature extraction. The generation of synthetic samples by perturbing existing data, for example, by adding

noise or occlusions, helps models to generalize better. Augmentation techniques, including rotation, translation, and scaling, are also effective. Creating synthetic pose data using physics-based simulations or generative models, such as GANs, can enrich the training dataset.

**Development of a framework to assure patient data privacy:** ML and, particularly, DL are increasingly becoming the dominant methods for uncovering knowledge in various industries. However, the effective deployment of data-driven applications requires access to extensive and diverse datasets. Yet, acquiring medical datasets poses significant challenges. Federated Learning (FL) addresses this hurdle by enabling collaborative learning without centralizing data, making it increasingly integrated into digital health applications. Data-driven machine learning (ML) emerges as a promising avenue for constructing precise and reliable statistical models, leveraging the vast trove of medical data collected by modern healthcare systems. Nonetheless, the limited utilization of this medical data in ML is largely due to its siloed nature and privacy concerns, which restrict data access. Without adequate data access, ML cannot fully realize its potential or transition from research to clinical practice. Federated Learning (FL), as a learning paradigm, aims to alleviate data governance and privacy concerns by training algorithms collaboratively without the need to share the data itself. FL allows for the decentralized training of ML models using datasets hosted remotely, eliminating the necessity to aggregate data and compromise its security. FL presents a promising avenue for enhancing ML-based systems to better adhere to regulatory requirements, bolster trustworthiness, and uphold data sovereignty. Exploring various FL approaches can ensure data privacy in training models relevant to CP patient care.

**Tailor therapies for individual needs:** Cerebral palsy includes a wide range of deformities depending on the area of brain injury and the secondary abnormalities each patient suffers. The treatment plan for these patients requires an interdisciplinary approach and requires inputs from experts in different fields, such as neuro, ortho, speech therapists, occupational therapists, physiotherapists, *etc*. This may lead to a difference in opinions on the treatment plans to be followed. AI-based quantification systems can be developed to assess the patient's health condition and plan the treatments for individual patients. Also, the analysis of patient progress over a period of time can be an added feature.

**Develop a method for screening infants for CP at birth:** Current methods of CP detection are less sensitive for preterm infants and newborns. This may lead to delays in the treatment interventions. Early identification is of great importance as the brain's plasticity is higher, and the treatment and plans can be worked out to decide the course of treatment. Research can be done on developing a method for screening premature babies and infants at birth.

## CONCLUSIONS

In summary, the amalgamation of artificial intelligence (AI) and machine learning methodologies presents considerable potential for diagnosing, predicting outcomes, and managing cerebral palsy (CP). By thoroughly reviewing existing literature, we delved into

the diverse applications of AI within healthcare, with a specific focus on CP treatment. Ranging from early detection to therapeutic interventions, AI-powered methods introduce innovative solutions capable of elevating the standard of care and enhancing patient outcomes. Our review has highlighted the diverse range of AI methods utilized in CP research, including machine learning algorithms, deep learning architectures, and federated learning paradigms. These approaches have demonstrated remarkable efficacy in analyzing medical data, extracting relevant features, and making accurate predictions regarding CP diagnosis and prognosis. Furthermore, we have identified key challenges and opportunities in the field, such as data privacy concerns, algorithm explainability, and the need for standardized evaluation metrics. Addressing these challenges will be essential for the widespread adoption and implementation of AI-driven solutions in clinical practice. Overall, our review underscores the transformative potential of AI in revolutionizing CP care. By leveraging advanced technologies and interdisciplinary collaborations, we can pave the way for more personalized, precise, and efficient healthcare interventions for individuals living with CP.

The integration of AI and ML techniques has revolutionized CP diagnosis. Researchers have explored various approaches, including automated GMA, image-based feature extraction, and genomic analysis. These methods aim to enhance early detection, improve accuracy, and streamline the diagnostic process. While AI and ML show promise, challenges remain, such as data quality, interpretability, and clinical adoption. Future research should focus on integrating multimodal data (clinical, imaging, genetic) and developing robust, interpretable models. Collaborations between clinicians, data scientists, and domain experts are essential to harness the full potential of AI and ML in CP diagnosis and personalized treatment planning.

### Funding
This research is funded by the Centre for Advanced Modelling and Geospatial Information Systems (CAMGIS), Faculty of Engineering and Information Technology, the University of Technology Sydney, Australia. This is also supported by the National Research Foundation of Korea (NRF) grant funded by the Korea government (Ministry of Science and ICT) (RS-2022-00165154, "Development of Application Support System for Satellite Information Big Data"). The funders had no role in study design, data collection and analysis, decision to publish, or preparation of the manuscript.

### Grant Disclosures
The following grant information was disclosed by the authors:
Centre for Advanced Modelling and Geospatial Information Systems (CAMGIS).
Faculty of Engineering and Information Technology.
University of Technology Sydney, Australia.
National Research Foundation of Korea (NRF): RS-2022-00165154.

## Competing Interests

The authors declare that they have no competing interests.

## Author Contributions

- Shalini Dhananjay Balgude conceived and designed the experiments, performed the experiments, analyzed the data, performed the computation work, prepared figures and/or tables, and approved the final draft.
- Shilpa Gite conceived and designed the experiments, performed the computation work, authored or reviewed drafts of the article, and approved the final draft.
- Biswajeet Pradhan conceived and designed the experiments, performed the computation work, authored or reviewed drafts of the article, and approved the final draft.
- Chang-Wook Lee performed the computation work, authored or reviewed drafts of the article, and approved the final draft.

## Data Availability

This is a literature review.

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
