# Peer review of "Artificial intelligence and machine learning approaches in cerebral palsy diagnosis, prognosis, and management: a comprehensive review"

_PeerJ Computer Science, doi:10.7717/peerj-cs.2505_

## Round 0.1 · original submission · Major Revisions

Based on the comments received, I suggest a major revision of the paper. Although the paper is well written and provides a thorough review of the literature on the application of AI and machine learning techniques in cerebral palsy (CP), there are several areas that need significant improvement. Some sections, especially the one on the drawbacks of current methods of CP care and management, need to be deepened to provide a more substantial contribution to the literature. Likewise, the method of calculating the mentioned evaluation metrics needs to be further detailed and the clarity of the figures and tables needs to be improved, with a special focus on Table 6.

Although the introductory section adequately highlights the importance of the topic, it would be useful to further refine it to improve its readability and understanding, especially for less experienced readers. The section on current methods of CP management needs further study and clarity. Furthermore, it is recommended to deepen the technical discussions on specific AI-ML algorithms and their comparative performance in CP applications. The inclusion of case studies or practical examples would enrich the paper, offering a more concrete view of successful implementations.

The limitations section of the study needs to be revised and enhanced, taking into account a broader perspective that includes aspects related to open access publishers. Additionally, it would be helpful to include a detailed roadmap or framework for integrating AI-ML into CP support. Finally, some figures, such as 1, 4, and 5, are difficult to read and should be replaced with higher resolution versions. Despite the strengths of the paper, it is essential to address these revisions to improve its impact and overall clarity.

**Language Note:** The review process has identified that the English language must be improved. PeerJ can provide language editing services - please contact us at [email protected] for pricing (be sure to provide your manuscript number and title). Alternatively, you should make your own arrangements to improve the language quality and provide details in your response letter. – PeerJ Staff

Reviewer 1 ·

Basic reporting

Professional English is used consistently and without ambiguity throughout.
The text contains adequate bibliographical references and provides sufficient foundational information and context on the field of study.
Figs. 1, 4 and 5 are difficult to read and have low resolution.

Experimental design

The methods are described correctly and clearly, the process and criteria are clear. The number of articles used is quite large.

Validity of the findings

The conclusions and future developments are encouraging and consistent with the research carried out.

·

Basic reporting

The paper is well-written, with professional language that is clear and engaging, but some chapters, especially "Introduction" and "Current Methods of Management and Treatment of CP", require further refinement to improve readability and understanding, especially for readers who are not very familiar with medical terminologies. The literature review is comprehensive, covering a wide range of AI and ML techniques applied to CP. The authors have cited sources adequately, providing a robust foundation for their review.

Experimental design

The paper could benefit from deeper technical discussions on specific AI-ML algorithms and their comparative performance in CP applications.

Additionally, it can be useful to include case studies or practical examples to illustrate successful implementations of AI-ML in CP diagnosis, prognosis, and management would be useful.

Validity of the findings

The paper effectively identifies significant shortcomings and challenges in current research, offering valuable insights for future studies. The discussion of ethical considerations and technical limitations adds depth to the review, highlighting the need for careful implementation of AI technologies in healthcare.

Additional comments

The conclusions are well-stated and linked to the original research questions, summarizing the benefits, challenges, and prospects of AI-ML in CP.

While the paper identifies future research directions, a more detailed roadmap or framework for integrating AI-ML into CP care could be beneficial. Overall, the paper presents a well-rounded review of AI and ML applications in CP, with a balanced discussion of benefits and challenges, and provides recommendations for future research to guide the field forward.

Reviewer 3 ·

Basic reporting

All comments have been added in detail to the last section.

Experimental design

All comments have been added in detail to the last section.

Validity of the findings

All comments have been added in detail to the last section.

Additional comments

Review Report for PeerJ Computer Science
(Artificial Intelligence and Machine Learning Approaches in Cerebral Palsy Diagnosis, Prognosis, and Management: A Comprehensive Review)

1. Within the scope of the study, studies in Cerebral Palsy in the literature related to machine learning and artificial intelligence are generally explained in detail.

2. The introduction section expresses the importance of the subject sufficiently.

3. When the Drawbacks of current methods of CP patient care section is examined in detail, it is observed that it is addressed from many different perspectives, but the sections in this section should definitely be detailed in order to contribute more to the literature.

4. The Research Goals and contributions of the study are clearly stated. However, the Limitations of the study section definitely needs to be reconsidered. It is recommended that the studies here be addressed from the perspective of open access publishers (PeerJ, MDPI, IOPScience, TechScience, Springer, etc.).

5. Although including evaluation metrics and their explanations are important, it is recommended that the calculation method of each metric mentioned be given in detail.

6. Even though the figures and tables seem appropriate in general, it is recommended that they be detailed, especially in Table-6, by adding columns such as "data preprocessing/augmentation, results".

In conclusion, although the article analyzes the literature in depth in terms of the topic discussed, all the sections itemized above should be taken into consideration.

---

## Round 0.2 · accepted · Accept

The authors have excellently fulfilled all the requests of the reviewers and helped to improve their paper.

Reviewer 1 ·

Basic reporting

Professional English is used consistently and without ambiguity throughout.
The text contains adequate bibliographical references and provides sufficient foundational information and context on the field of study.

Experimental design

The article falls within the objectives and scope of the journal and is suitable for the type of contribution requested. The research was conducted rigorously, adhering to high technical and ethical standards.

The survey methodology ensures comprehensive and unbiased coverage of the topic. Sources are properly cited, with correct use of quotations.

Validity of the findings

The conclusions and potential future developments are promising and align well with the research conducted.

Reviewer 3 ·

Basic reporting

All comments have been added in detail to the last section.

Experimental design

All comments have been added in detail to the last section.

Validity of the findings

All comments have been added in detail to the last section.

Additional comments

Review Report for PeerJ Computer Science
(Artificial Intelligence and Machine Learning Approaches in Cerebral Palsy Diagnosis, Prognosis, and Management: A Comprehensive Review)

Thanks for the revision. The changes made to the paper are sufficient. I recommend that the paper be accepted. Best regards.